# ProtoDiff: Learning to Learn Prototypical Networks by Task-Guided Diffusion

**Yingjun Du[1], Zehao Xiao[1], Shencai Liao[2] , Cees G. M. Snoek[1]**
[1]AIM Lab, University of Amsterdam [2]Inception Institute of Artificial Intelligence

## Abstract

Prototype-based meta-learning has emerged as a powerful technique for addressing few-shot learning challenges. However, estimating a deterministic prototype using a simple average function from a limited number of examples remains a fragile process. To overcome this limitation, we introduce ProtoDiff, a novel framework that leverages a task-guided diffusion model during the meta-training phase to gradually generate prototypes, thereby providing efficient class representations. Specifically, a set of prototypes is optimized to achieve per-task prototype overfitting, enabling accurately obtaining the overfitted prototypes for individual tasks. Furthermore, we introduce a task-guided diffusion process within the prototype space, enabling the meta-learning of a generative process that transitions from a vanilla prototype to an overfitted prototype. ProtoDiff gradually generates task-specific prototypes from random noise during the meta-test stage, conditioned on the limited samples available for the new task. Furthermore, to expedite training and enhance ProtoDiff's performance, we propose the utilization of residual prototype learning, which leverages the sparsity of the residual prototype. We conduct thorough ablation studies to demonstrate its ability to accurately capture the underlying prototype distribution and enhance generalization. The new state-of-the-art performance on within-domain, cross-domain, and few-task few-shot classification further substantiates the benefit of ProtoDiff.

## 1 Introduction

This paper considers prototype-based meta-learning, where models are trained to swiftly adapt to current tasks and perform classification through metric-based comparisons between examples and newly introduced variables - the prototypes of the classes. This approach, rooted in works by Reed [34] and further developed by Snell *et al.*[42], generalizes deep learning models to scenarios where labeled data is scarce. The fundamental idea is to compute the distances between queried examples and class prototypes and perform classification based on these distances to predict classes for queried examples. Derived from the prototypical network [42], several prototype-based methods have demonstrated their effectiveness in few-shot learning [2, 8, 11, 61]. However, as prototypes are estimated from a limited number of sampled examples, they may not accurately capture the overall distribution [54, 61]. Such biased distributions could lead to biased prototypes and subsequent classification errors, suggesting that the current prototype modeling approach might lack the ability to represent universal class-level information effectively. Rather than relying on points in the prototype embedding space by using a simple average function, we propose to generate the distribution of prototypes for each task.

To overcome the challenges of estimating deterministic prototypes from limited examples in few-shot learning, we propose ProtoDiff, an innovative framework that leverages a task-guided diffusion model during the meta-training phase. ProtoDiff introduces a three-step process to address these limitations comprehensively. Firstly, we optimize a set of prototypes to achieve per-task prototype overfitting,

37th Conference on Neural Information Processing Systems (NeurIPS 2023).

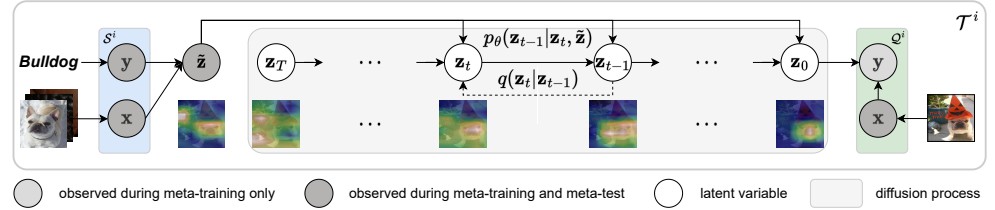

**Figure 1: Computational graph of ProtoDiff.** In the initial stage, the computation of the vanilla prototype $\tilde{\mathbf{z}}$ is performed based on the support set $\mathcal{S}^i$. Subsequently, through the process of diffusion sampling, the diffused prototype $\mathbf{z}_{t-1}$ is derived from the combination of $\mathbf{z}_t$ and $\tilde{\mathbf{z}}$. Finally, the prediction $\mathbf{y}$ for the query set $\mathcal{Q}^i$ is generated by utilizing the diffused prototype $\mathbf{z}_0$ in conjunction with the query set $\mathbf{x}$. The diffusion forward and sampling processes are indicated by dashed and solid arrows within the gray rectangular area.

ensuring accurate retrieval of overfitted prototypes specific to individual tasks. These overfitted prototypes serve as the *ground truth* representations for each task. Secondly, a task-guided diffusion process is implemented within the prototype space, enabling meta-learning of a generative process that smoothly transitions prototypes from their vanilla form to the overfitted state. ProtoDiff generates task-specific prototypes during the meta-test stage by conditioning random noise on the limited samples available for the given new task. Finally, we propose residual prototype learning, which significantly accelerates training and further enhances the performance of ProtoDiff by leveraging the sparsity of residual prototypes. The computational graph of ProtoDiff, illustrated in Figure 1, showcases the sequential steps involved in the forward diffusion process on the prototype and the generative diffusion process. The resulting generated prototype, denoted as $\mathbf{z}_0$, is utilized for query prediction. By incorporating probabilistic and task-guided prototypes, ProtoDiff balances adaptability and informativeness, positioning it as a promising approach for augmenting few-shot learning in prototype-based meta-learning models.

To validate the effectiveness of ProtoDiff, we conduct comprehensive experiments on three distinct few-shot learning scenarios: within-domain, cross-domain, and few-task few-shot learning. Our findings reveal that ProtoDiff significantly outperforms state-of-the-art prototype-based meta-learning models, underscoring the potential of task-guided diffusion to boost few-shot learning performance. Furthermore, we provide a detailed analysis of the diffusion mechanism employed in ProtoDiff, showcasing its ability to capture the underlying data structure better and improve generalization. This thorough investigation highlights the strengths of our approach, demonstrating its potential to offer a more effective solution for few-shot learning tasks in various applications and settings.

## 2 Preliminaries

Before detailing our ProtoDiff methodology, we first present the relevant background on few-shot classification, the prototypical network, and diffusion models.

**Few-shot classification.** We define the N-way K-shot classification problem, which consists of support sets $\mathcal{S}$ and a query set $\mathcal{Q}$. Each task $\mathcal{T}^i$, also known as an episode, represents a classification problem sampled from a task distribution $p(\mathcal{T})$. The *way* of an episode denotes the number of classes within the support sets, while the *shot* refers to the number of examples per class. Tasks are created from a dataset by randomly selecting a subset of classes, sampling points from these classes, and subsequently dividing the points into support and query sets. The episodic optimization approach [50] trains the model iteratively, performing one episode update at a time.

**Prototypical network.** We develop our method based on the prototypical network (ProtoNet) by Snell *et al.* [42]. Specifically, the ProtoNet leverages a non-parametric classifier that assigns a query point to the class having the nearest prototype in the learned embedding space. The prototype $\mathbf{z}^c$ of an object class $c$ is obtained by: $\mathbf{z}^c = \frac{1}{K} \sum_k f_\phi(\mathbf{x}^{c,k})$, where $f_\phi(\mathbf{x}^{c,k})$ is the feature embedding of the support sample $\mathbf{x}^{c,k}$, which is usually obtained by a convolutional neural network. For each query sample $\mathbf{x}^q$, the distribution over classes is calculated based on the softmax over distances to the prototypes of all classes in the embedding space:

$$p(\mathbf{y}_n^q = c | \mathbf{x}^q) = \frac{\exp(-d(f_\phi(\mathbf{x}^q), \mathbf{z}^c))}{\sum_{c'} \exp(-d(f_\phi(\mathbf{x}^q), \mathbf{z}^{c'}))}, \tag{1}$$

where $\mathbf{y}^q$ denotes a random one-hot vector, with $\mathbf{y}_n^q$ indicating its $n$-th element, and $d(\cdot, \cdot)$ is some (Euclidean) distance function. Due to its non-parametric nature, the ProtoNet enjoys high flexibility and efficiency, achieving considerable success in few-shot learning. To avoid confusion, we omit the superscript $c$ for the prototype $\mathbf{z}$ in this subsection.

**Diffusion model.** In denoising diffusion probabilistic models [16], a forward diffusion process, $q(\boldsymbol{x}_t|\boldsymbol{x}_{t-1})$, is characterized as a Markov chain that progressively introduces Gaussian noise at each time step $t$, beginning with a clean image $\boldsymbol{x}_0 \sim q(\boldsymbol{x}_0)$. The subsequent forward diffusion process is formulated as follows:

$$q(\boldsymbol{x}_T|\boldsymbol{x}_0) := \prod_{t=1}^{T} q(\boldsymbol{x}_t|\boldsymbol{x}_{t-1}), \quad \text{where} \quad q(\boldsymbol{x}_t|\boldsymbol{x}_{t-1}) := \mathcal{N}(\boldsymbol{x}_t; \sqrt{1-\beta_t}\boldsymbol{x}_{t-1}, \beta_t \boldsymbol{I}), \quad (2)$$

where $\{\beta\}_{t=0}^{T}$ is a variance schedule. By defining $\alpha_t := 1 - \beta_t$ and $\bar{\alpha}_t := \prod_{s=1}^{t} \alpha_s$, the forward diffused sample at time step $t$, denoted as $\boldsymbol{x}_t$, can be generated in a single step as follows:

$$\boldsymbol{x}_t = \sqrt{\bar{\alpha}_t}\boldsymbol{x}_0 + \sqrt{1-\bar{\alpha}_t}\boldsymbol{\epsilon}, \quad \text{where} \quad \boldsymbol{\epsilon} \sim \mathcal{N}(\mathbf{0}, \boldsymbol{I}). \quad (3)$$

Since the reverse of the forward step, $q(\boldsymbol{x}_{t-1}|\boldsymbol{x}_t)$, is computationally infeasible, the model learns to maximize the variational lower bound using parameterized Gaussian transitions, $p_\theta(\boldsymbol{x}_{t-1}|\boldsymbol{x}_t)$, where the parameter is denoted as $\theta$. Consequently, the reverse process is approximated as a Markov chain with the learned mean and fixed variance, starting from a random noise $\boldsymbol{x}_T \sim \mathcal{N}(\boldsymbol{x}_T; \mathbf{0}, \boldsymbol{I})$:

$$p_\theta(\boldsymbol{x}_{0:T}) := p_\theta(\boldsymbol{x}_T) \prod_{t=1}^{T} p_\theta(\boldsymbol{x}_{t-1}|\boldsymbol{x}_t), \quad (4)$$

where

$$p_\theta(\boldsymbol{x}_{t-1}|\boldsymbol{x}_t) := \mathcal{N}(\boldsymbol{x}_{t-1}; \boldsymbol{\mu}_\theta(\boldsymbol{x}_t, t), \sigma_t^2 \boldsymbol{I}), \quad \boldsymbol{\mu}_\theta(\boldsymbol{x}_t, t) := \frac{1}{\sqrt{\alpha_t}}\left(\boldsymbol{x}_t - \frac{1-\alpha_t}{\sqrt{1-\bar{\alpha}_t}}\boldsymbol{\epsilon}_\theta(\boldsymbol{x}_t, t)\right). \quad (5)$$

Here, $\boldsymbol{\epsilon}_\theta(\boldsymbol{x}_t, t)$ is the diffusion model trained by optimizing the following objective function:

$$\mathcal{L}_\theta = \mathbb{E}_{t, \boldsymbol{x}_0, \boldsymbol{\epsilon}}\left[\|\boldsymbol{\epsilon} - \boldsymbol{\epsilon}_\theta(\sqrt{\bar{\alpha}_t}\boldsymbol{x}_0 + \sqrt{1-\bar{\alpha}_t}\boldsymbol{\epsilon}, t)\|^2\right]. \quad (6)$$

Upon completing the optimization, the learned score function is integrated into the generative (or reverse) diffusion process. To sample from $p_\theta(\boldsymbol{x}_{t-1}|\boldsymbol{x}_t)$, one can perform the following:

$$\boldsymbol{x}_{t-1} = \boldsymbol{\mu}_\theta(\boldsymbol{x}_t, t) + \sigma_t \boldsymbol{\epsilon} = \frac{1}{\sqrt{\alpha_t}}\left(\boldsymbol{x}_t - \frac{1-\alpha_t}{\sqrt{1-\bar{\alpha}_t}}\boldsymbol{\epsilon}_\theta(\boldsymbol{x}_t, t)\right) + \sigma_t \boldsymbol{\epsilon}. \quad (7)$$

In the case of conditional diffusion models [22, 38, 39], the diffusion model $\boldsymbol{\epsilon}_\theta$ in equations (6) and (7) is substituted with $\boldsymbol{\epsilon}_\theta(\mathbf{c}, \sqrt{\bar{\alpha}_t}\boldsymbol{x}_0 + \sqrt{1-\bar{\alpha}_t}\boldsymbol{\epsilon}, t)$, where $\mathbf{c}$ represents the corresponding conditions, e.g., other images, languages, and sounds, etc. Consequently, the matched conditions strictly regulate the sample generation in a supervised manner, ensuring minimal changes to the image content. Dhariwal and Nichol [7] suggested classifier-guided image translation to generate the images of the specific classes along with a pre-trained classifier. Taking inspiration from conditional diffusion models that excel at generating specific images using additional information, we introduce the task-guided diffusion model that generates prototypes of specific classes by taking into account contextual information from various few-shot tasks. Note that our task-guided diffusion model operates not on an image $\boldsymbol{x}$, but on the prototype $\mathbf{z}$.

## 3 Methodology

This section outlines our approach to training prototypical networks via task-guided diffusion. We begin by explaining how to obtain task-specific overfitted prototypes in Section 3.1. Next, we introduce the task-guided diffusion method for obtaining diffused prototypes in Section 3.2. In the same section, we also introduce residual prototype learning, which accelerates training. Figure 2 visualizes the diffusion process of diffused prototypes by our ProtoDiff.

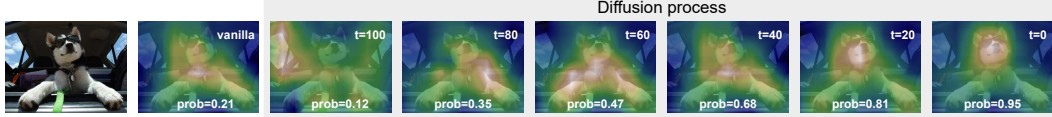

**Figure 2: Visualization of the diffusion process.** ProtoDiff randomly selects certain areas to predict during the diffusion process, with the lowest probability at the beginning time step. As time progresses, the prototype gradually aggregates towards the *dog*, with the highest probability at t=0.

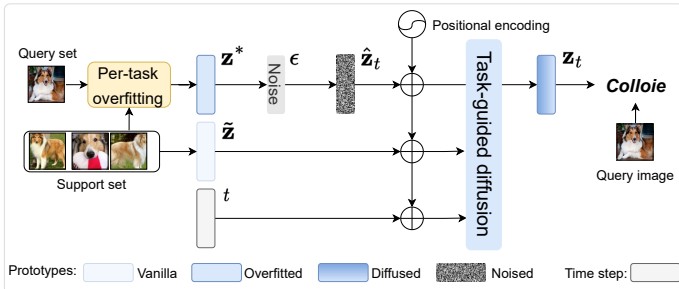

**Figure 3: ProtoDiff illustration.** We first obtain an overfitted prototype by the per-task prototype overfitting. We then add noise to the overfitted prototype, which inputs the diffusion forward process. The input of task-guided diffusion includes the vanilla prototype $\tilde{\mathbf{z}}$ and a random time step $t$. The resulting output is the diffused prototype $\mathbf{z}_t$.

## 3.1 Per-task prototype overfitting

Our approach utilizes a modified diffusion process as its foundation. Firstly, a meta-learner, denoted by $f_\phi(\mathcal{T}^i) = f(\mathcal{T}^i, \phi)$, is trained on the complete set of tasks, $\mathcal{T}$. Here, $\phi$ corresponds to the learned weights of the model over the entire task set. Subsequently, fine-tuning is performed on each individual task $\mathcal{T}^i$ to obtain task-specific overfitted prototypes denoted by $\mathbf{z}^{i,*} = f\phi^i(\mathcal{S}^i, \mathcal{Q}^i)$. This involves running the meta-learner $f_\phi$ on the support set $\mathcal{S}^i$ and a query set $\mathcal{Q}^i$ of the task $\mathcal{T}^i$.

$$\phi^i = \phi - \eta \sum_{(\mathbf{x},\mathbf{y}) \sim \mathcal{T}_i}^{\mathcal{T}_i} \mathcal{L}_{\mathrm{CE}}(f(\mathcal{S}^i, \mathbf{x}^{q^i}, \phi), \mathbf{y}^{q^i}), \tag{8}$$

where $\mathcal{L}_{\mathrm{CE}}$ is the cross-entropy loss minimized in the meta-learner training. The support set $\mathcal{S}^i$ and query set $\mathcal{Q}^i = \{\mathbf{x}^{q^i}, \mathbf{y}^{q^i}\}$ correspond to the data used for fine-tuning task $\mathcal{T}^i$. During fine-tuning, we obtain the $\phi^i$ through several iterations of gradient descent. We illustrate the overall framework of the per-task prototype learning in the appendix.

Our proposed method, ProtoDiff, for few-shot learning, relies on acquiring task-specific overfitted prototypes that can be considered as the "optima" prototypes due to their high confidence in final predictions, approaching a value of 1. To achieve this, we employ fine-tuning of the meta-learner and extract task-specific information that surpasses the accuracy and reliability of generic prototypes used in the meta-training stage. However, accessing the query set in the meta-test stage is not feasible. Hence, we need to rely on the vanilla prototype to meta-learn the process of generating the overfitted prototype during the meta-training stage. In the forthcoming section, we will introduce the task-guided diffusion model, which facilitates the learning process transitioning from the vanilla prototype to the overfitted prototype.

## 3.2 Task-guided diffusion

Our generative overfitted prototype is based on diffusion [43], a robust framework for modeling the prototypes instead of images. The length of the diffusion process $T$ determines the number of forward passes needed to generate a new prototype, rather than the dimensionality of the data. Our model uses diffusion to progressively denoise the overfitted prototype $\mathbf{z}^*$.

**Meta-training phase.** Diffusion models can be configured to predict either the signal or the noise when presented with a noisy input [16, 31]. Previous research in the image domain has suggested that noise prediction is superior to signal prediction. However, we discovered empirically that signal prediction performs better than noise prediction in our experiments, so we parameterized the diffusion model to output the prototype.

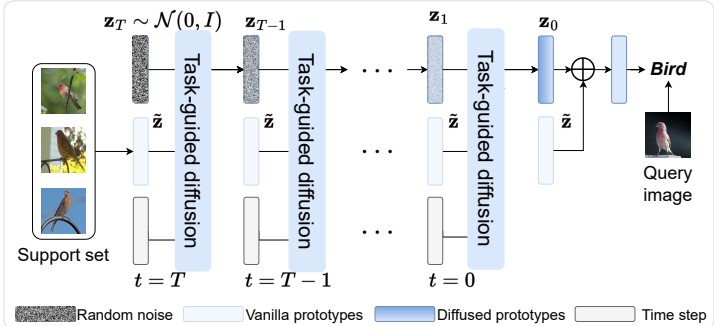

Figure 4: **Diffusion sampling during meta-test.** Sampling starts from a random noise $\mathbf{z_T}$ and gradually denoises it to the diffused prototypes $\mathbf{z_t}$. At each time step $t$ is sampled by taking the $\mathbf{z_{t+1}}$, vanilla prototypes $\tilde{\mathbf{z}}$, and time step $t$ as inputs. The diffused prototypes $\mathbf{z_0}$ are used to predict the query set.

The generative diffusion process is devised to reconstruct the overfitted prototype $\mathbf{z}^*$ by iteratively operating on a random noise vector $\mathbf{z}_T \sim \mathcal{N}(\mathbf{0}, \boldsymbol{I})$, which possesses the same dimensions as $\mathbf{z}^*$. This process continues with $\mathbf{z}_t$, where $t$ denotes the number of diffusion steps. Consequently, the reconstructed $\mathbf{z}_0$ should exhibit proximity to $\mathbf{z}^*$ for a given task.

Specifically, during the forward diffusion process at time step $t$, we obtain the noised overfitted prototype $\hat{\mathbf{z}}_t$. Subsequently, we input the noised prototype $\hat{\mathbf{z}}_t$, the vanilla prototype $\tilde{\mathbf{z}}$, and the time step $t$ into the task-guided diffusion. This yields the diffused prototype $\mathbf{z}_t$, which then allows us to predict the final results of the query set using Equation (1). For each task, our task-guided diffusion entails two components: the variational lower bound $\mathcal{L}_{\mathrm{diff}}$ for the diffused prototype, and the cross-entropy loss $\mathcal{L}_{\mathrm{CE}}$.

The objective is to minimize the simplified variational lower bound, which involves predicting the denoised overfitted prototype:

$$\mathcal{L}_{\mathrm{diff}} = |\mathbf{z}^* - \mathbf{z}_\theta(\sqrt{\bar{\alpha}_t}\mathbf{z}^* + \sqrt{1 - \bar{\alpha}_t}\boldsymbol{\epsilon}, \tilde{\mathbf{z}}, t)|^2, \tag{9}$$

Here, $\mathbf{z}_\theta(\cdot, \cdot)$ is implemented by the transformer model [49] operating on prototype tokens from $\mathbf{z}^*$, $\tilde{\mathbf{z}}$, and the time step.

By utilizing equation (1), we derive the final prediction $\hat{\mathbf{y}}^q$ using the diffused prototype $\mathbf{z}_t$. The ultimate objective is expressed as follows:

$$\mathcal{L} = \sum_{(\mathbf{x},\mathbf{y})\sim\mathcal{Q}}^{|Q|} \Big[ -\mathbb{E}q(\mathbf{z}_t|\mathbf{z}_{t+1}, \tilde{\mathbf{z}})\big[\log p(\mathbf{y}^q|\mathbf{x}^q, \mathbf{z}_t)\big] + \beta|\mathbf{z}^* - \mathbf{z}_\theta(\sqrt{\bar{\alpha}_t}\mathbf{z}^* + \sqrt{1 - \bar{\alpha}_t}\boldsymbol{\epsilon}, \tilde{\mathbf{z}}, t)|^2 \Big], \tag{10}$$

where $\beta$ represents a hyperparameter, and $|Q|$ denotes the query size.

To prepare the two input prototypes $\mathbf{z}^*$ and $\tilde{\mathbf{z}}$ for processing by the transformer, we assign each position token inspired by [10]. We also provide the scalar input diffusion timestep $t$ as individual tokens to the transformer. To represent each scalar as a vector, we use a frequency-based encoding scheme [28]. Our transformer architecture is based on GPT-2 [33]. The decoder in the final layer of our ProtoDiff maps the transformer's output to the diffused prototype. Note that only the output tokens for the noised overfitted prototypes $\hat{\mathbf{z}}_t$ are decoded to predictions. The overall meta-training phase of ProtoDiff is shown in Figure 3.

**Meta-test phase.** We can not obtain the overfitted prototype during the meta-test phase since we can not access the query set. Thus, diffusion sampling begins by feeding-in Gaussian noise $\mathbf{z}_T$ as the $\hat{\mathbf{z}}$ input and gradually denoising it. Specifically, to handle a new task $\tau = \{\mathcal{S}, \mathcal{Q}\}$, we first compute the vanilla prototype $\tilde{\mathbf{z}}$ using the support set $\mathcal{S}$. We then randomly sample noise $\epsilon$ from $\mathcal{N}(\mathbf{0}, \boldsymbol{I})$. We input both $\tilde{\mathbf{z}}$ and $\epsilon$ to the learned task-guided diffusion model to obtain the diffused prototype $\mathbf{z}_{T-1} = \mathbf{z}_\theta(\mathbf{z}_T, \tilde{\mathbf{z}}, T)$. After $T$ iterations, the final diffused prototype can be obtained as $\mathbf{z}_0 = \mathbf{z}_\theta(\mathbf{z}_1, \mathbf{z}, t_0)$. Once the final diffused prototype $\mathbf{z}_0$ is obtained, we calculate the final prediction $\hat{\mathbf{y}}^q$ using equation (1). Figure 4 illustrates sampling in the meta-test stage. We also provide the detailed algorithm of the meta-training and meta-test phase in the appendix.

**Residual prototype learning.** To further enhance the performance of ProtoDiff, we introduce a residual prototype learning mechanism in our framework. We observe that the differences between the overfitted prototype $\mathbf{z}^*$ and its vanilla prototype $\tilde{\mathbf{z}}$ are not significant, as the vector $\mathbf{z}^* - \tilde{\mathbf{z}}$ contains many zeros. Therefore, we propose to predict the prototype update $\mathbf{z}^* - \tilde{\mathbf{z}}$ instead of directly predicting

the overfitted prototype $\mathbf{z}^*$ itself. This approach also enables us to initialize ProtoDiff to perform the identity function by setting the decoder weights to zero. Moreover, we find that the global residual connection, combined with the identity initialization, significantly speeds up training. By utilizing this mechanism, we improve the performance of ProtoDiff in few-shot learning tasks.

## 4 Related Work

**Prototype-based meta-learning.** Prototype-based meta-learning is based on distance metrics and generally learns a shared or adaptive embedding space in which query images are accurately matched to support images for classification. It relies on the assumption that a common metric space is shared across related tasks and usually does not employ an explicit base learner for each task. By extending the matching network [50] to few-shot scenarios, Snell *et al.* [42] constructed a prototype for each class by averaging the feature representations of samples from the class in the metric space. The classification matches the query samples to prototypes by computing their distances. To enhance the prototype representation, Allen et al [2] proposed an infinite mixture of prototypes to adaptively represent data distributions for each class, using multiple clusters instead of a single vector. Oreshkin et al [32] proposed a task-dependent adaptive metric for few-shot learning and established prototype classes conditioned on a task representation encoded by a task embedding network. Yoon et al [60] proposed a few-shot learning algorithm aided by a linear transformer that performs task-specific null-space projection of the network output. Graphical neural network-based models generalize the matching methods by learning the message propagation from the support set and transferring it to the query set [13]. FEAT [58] was proposed to leverages samples from all categories within a task to generate a prototype, capitalizing on the intrinsic inter-class information to derive a more discriminative prototype. In contrast, our ProtoDiff, while using the overfitted prototype as supervision, only employs samples from a single category to generate the new prototype. This means we are not tapping into the potential informative context provided by samples from other categories. Additionally, our ProtoDiff employs the diffusion model to progressively generate the prototype, whereas FEAT does not utilize any generative model for prototype creation. Prototype-based methods have recently been improved in various ways [4, 46, 58, 61]. To the best of our knowledge, we are the first to propose a diffusion method to generate the prototype per task, rather than a deterministic function.

**Diffusion models.** These models belong to a category of neural generative models that utilize stochastic diffusion processes [43, 44], much like those found in thermodynamics. In this framework, a gradual introduction of noise is applied to a sample of data. A neural model then learns to reverse the process by progressively removing noise from the sample. The model denoises an initial pure noise to obtain samples from the learned data distribution. Ho *et al.* [16] and Song *et al.* [44] contributed to advancements in image generation, while Dhariwal and Nichol [7] introduced classifier-guided diffusion for a conditioned generation. GLIDE later adapted this approach [31], enabling conditioning on textual CLIP representations. Classifier-free guidance [17] allows for conditioning with a balance between fidelity and diversity, resulting in improved performance [31]. Since the guided diffusion model requires a large number of image-annotation pairs for training, Hu *et al.* [19] propose self-guided diffusion models. Recently, Hyperdiffusion [12, 27] was proposed in the weight space for generating implicit neural fields and 3D reconstruction. In this paper, we introduce ProtoDiff, a prototype-based meta-learning approach within a task-guided diffusion model that incrementally improves the prototype's expressiveness by utilizing a limited number of samples.

## 5 Experiments

In this section, we assess the efficacy of ProtoDiff in the context of three distinct few-shot learning scenarios: within-domain few-shot learning, cross-domain few-shot learning, and few-task few-shot learning. For the within-domain few-shot learning experiments, we apply our method to three specific datasets: *mini*Imagenet [50], *tiered*Imagenet [35], and ImageNet-800 [5]. Regarding cross-domain few-shot learning, we utilize *mini*Imagenet [50] as the training domain, while testing is conducted on four distinct domains: CropDisease [30], EuroSAT [15], ISIC2018 [47], and ChestX [52]. Furthermore, few-task few-shot learning [56] challenges the common assumption of having abundant tasks available during meta-training. To explore this scenario, we perform experiments on four few-task meta-learning challenges: *mini*Imagenet-S [50], ISIC [29], DermNet-S [6], and Tabular

**Table 1: Benefit of ProtoDiff** on *mini*Imagenet, *tiered*Imagenet and ImageNet-800. The results of the Classifier-Baseline and Meta-Baseline are provided by Chen *et al.* [5]. ProtoDiff consistently achieves better performance than two baselines on all datasets and settings.

| Method | *mini*Imagenet | | *tiered*Imagenet | | Imagenet-800 | |
|---|---|---|---|---|---|---|
| | 1-shot | 5-shot | 1-shot | 5-shot | 1-shot | 5-shot |
| Classifier-Baseline [5] | $58.91_{\pm0.23}$ | $77.76_{\pm0.17}$ | $68.07_{\pm0.29}$ | $83.74_{\pm0.18}$ | $86.07_{\pm0.21}$ | $96.14_{\pm0.08}$ |
| Meta-Baseline [5] | $63.17_{\pm0.23}$ | $79.26_{\pm0.17}$ | $68.62_{\pm0.27}$ | $83.74_{\pm0.18}$ | $89.70_{\pm0.19}$ | $96.14_{\pm0.08}$ |
| **ProtoDiff** | $\mathbf{66.63}_{\pm0.21}$ | $\mathbf{83.48}_{\pm0.15}$ | $\mathbf{72.95}_{\pm0.24}$ | $\mathbf{85.15}_{\pm0.18}$ | $\mathbf{92.13}_{\pm0.20}$ | $\mathbf{98.21}_{\pm0.08}$ |

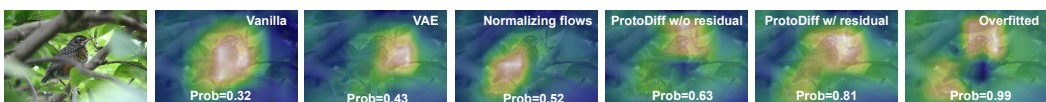

**Figure 5: Visualization of the different reconstructed overfitted prototypes using different generative models.** The vanilla prototype focuses only on the overall features of the bird and overlooks the finer details of the *robin*. In contrast, the overfitted prototype can highlight specific features such as the tail and beak. While VAE and normalizing flow can generate prototypes of certain parts, the diffusion method can generate prototypes that more closely resemble the overfitted prototype. With residual prototype learning, ProtoDiff achieves better.

Murris [3]. For a comprehensive understanding of the datasets used in each setting, we provide detailed descriptions in the Appendix.

**Implementation details.** In our within-domain experiments, we utilize a Conv-4 and ResNet-12 backbone for *mini*Imagenet and *tiered*Imagenet. A ResNet-50 is used for ImageNet-800. We follow the approach described in [5] to achieve better performance and initially train a feature extractor on all the meta-training data without episodic training. Standard data augmentation techniques are applied, including random resized crop and horizontal flip. For our cross-domain experiments, we use a ResNet-10 backbone to extract image features, which is a common choice for cross-domain few-shot classification [14, 57]. The training configuration for this experiment is the same as the within-domain training. For few-task few-shot learning, we follow [56] using a network containing four convolutional blocks and a classifier layer. The average within-domain/ cross-domain, few-task few-shot classification accuracy (%, top-1) along with 95% confidence intervals are reported across all test query sets and tasks. Code available at: https://github.com/YDU-uva/ProtoDiff.

**Benefit of ProtoDiff.** Table 1 compares our ProtoDiff with two baselines to demonstrate its effectiveness. The first Classifier-Baseline is a classification model trained on the entire label set using a classification loss and performing few-shot tasks with the cosine nearest-centroid method. The Meta-Baseline [5] consists of two stages. In the first stage, a classifier is trained on all base classes, and the last fully connected layer is removed to obtain the feature encoder. The second stage is meta-learning, where the classifier is optimized using episodic training. The comparison between the Baseline and Meta-Baseline highlights the importance of episodic training for few-shot learning. ProtoDiff consistently outperforms Meta-Baseline [5] by a large margin on all datasets and shots. The task-guided diffusion model employed in our ProtoDiff generates more informative prototypes, leading to improvements over deterministic prototypes.

**Benefit of diffusion model.** To confirm that the performance gain of our ProtoDiff model can be attributed to the diffusion model, we conducted the experiments only using MLP and transformers as non-generative models. We also compared it with two widely used generative models: the variational autoencoder (VAE) [21] and normalizing flows [36]. VAE learns a low-dimensional representation of input data to generate new samples. Normalizing flows are more recent and learn a sequence of invertible transformations to map a simple prior distribution to the data distribution. We obtained the task-specific prototype in the meta-test stage, conditioned on the support samples and $\epsilon \sim \mathcal{N}(0, I)$, by first acquiring an overfit-

**Table 2: Benefit of diffusion model** over (non-)generative models on *mini*Imagenet.

| Method | *mini*Imagenet | |
|---|---|---|
| | 1-shot | 5-shot |
| w/o generative model [5] | $63.17_{\pm0.23}$ | $79.26_{\pm0.17}$ |
| w/ MLP | $64.15_{\pm0.21}$ | $80.23_{\pm0.15}$ |
| w/ Transformer | $64.97_{\pm0.21}$ | $81.28_{\pm0.14}$ |
| w/ VAE | $64.45_{\pm0.22}$ | $80.13_{\pm0.15}$ |
| w/ Normalizing flows | $65.11_{\pm0.22}$ | $81.96_{\pm0.17}$ |
| **w/ Diffusion** | $\mathbf{66.63}_{\pm0.21}$ | $\mathbf{83.48}_{\pm0.15}$ |

ted prototype $\mathbf{z}^*$ and then using it as the ground truth to train VAE and normalizing flows in the meta-training stage. The experimental results reported in Table 2 show that the diffusion model outperforms all variants in terms of accuracy. Specifically, our diffusion model improves accuracy by 2.18% compared to VAE and 1.52% compared to normalizing flows. Furthermore, we reconstructed the overfitted prototype using different generative models in Figure 5. The vanilla prototype focuses solely on the bird's features and overlooks the finer details of the *robin*. In contrast, the overfitted prototype can emphasize specific features such as the tail and beak, resulting in better discrimination of the *robin* from other bird species. While VAE and normalizing flows can generate prototypes of specific parts, our diffusion method can generate prototypes that more closely resemble the overfitted prototype, leading to improved performance. Our findings suggest that diffusion models hold promise as a more practical approach for few-shot learning, as they can model complex distributions and produce informative prototypes.

**Effect of the residual prototype.** The incorporation of residual prototypes in ProtoDiff presents a viable approach for reducing computational costs. This is attributable to the fact that the residual prototype exclusively encapsulates the disparity between the overfitted prototype and the original prototype, while the latter can be readily reused in each episode. Consequently, the calculation of the overfitted prototype is only necessitated once during the meta-training stage. Notably, the residual prototype often encompasses numerous zero values, thereby further expediting the training process. Table 6a illustrates the superior accuracy achieved by ProtoDiff with residual prototype learning in comparison to both the vanilla model and ProtoDiff without the residual prototype. The training progress of ProtoDiff is visually represented in Figure 6b, demonstrating its accelerated training capabilities when compared to alternative methods. These results indicate the effectiveness of residual prototype learning in capturing task-specific information previously unaccounted for by the vanilla prototypes. Thus, the inclusion of residual prototype learning in ProtoDiff not only expedites the training process but also serves as a straightforward and effective approach to enhance the overall performance of ProtoDiff.

**(a)** Accuracy (%) on various settings.

| Method | *mini*Imagenet | |
| | 1-shot | 5-shot |
| --- | --- | --- |
| Vanilla | $63.17_{\pm 0.23}$ | $79.26_{\pm 0.17}$ |
| ProtoDiff w/o residual | $64.75_{\pm 0.22}$ | $80.76_{\pm 0.16}$ |
| **ProtoDiff w/ residual** | $\mathbf{66.63_{\pm 0.21}}$ | $\mathbf{83.48_{\pm 0.15}}$ |

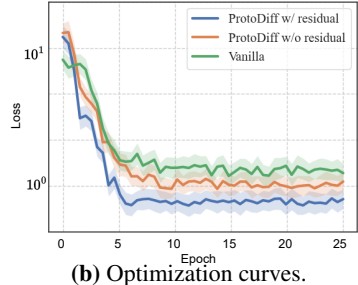

**(b)** Optimization curves.

**Figure 6: Effect of residual prototype** on *mini*Imagenet. Utilizing ProtoDiff with a residual prototype not only accelerates the training process but also surpasses the performance of both the vanilla prototype and the non-residual prototype.

**Analysis of uncertainty.** In the process of estimating the log-likelihood of input data, the choice of the number of Monte Carlo samples becomes a critical hyperparameter. Typically, a larger number of samples tends to provide a more accurate approximation and better generalization to unforeseen prompts. In our experiments, we employed Meta-Baseline and FEAT [58] with ProtoDiff, incorporating the sampling of multiple prototypes in the final phase of the diffusion process. The results of these experiments are provided in Table 3. Notably, we observe a substantial performance boost with an increase in the number of samples. For instance, a sample size of 50 results in a significant enhancement in 1-shot accuracy, raising it from 66.63 for Meta-Baseline + ProtoDiff to 72.25 for FEAT + ProtoDiff.

**Table 3: Analysis of uncertainty.** Increasing the number of Monte Carlo samples provides a good improvement on *mini*Imagenet.

| | Meta-baseline + ProtoDiff | | FEAT + ProtoDiff | |
| | 1-shot | 5-shot | 1-shot | 5-shot |
| --- | --- | --- | --- | --- |
| 1 | $66.63_{\pm 0.21}$ | $83.48_{\pm 0.15}$ | $68.97_{\pm 0.25}$ | $85.16_{\pm 0.17}$ |
| 10 | $68.02_{\pm 0.17}$ | $84.95_{\pm 0.10}$ | $70.18_{\pm 0.15}$ | $86.53_{\pm 0.12}$ |
| 20 | $68.91_{\pm 0.15}$ | $85.74_{\pm 0.10}$ | $61.07_{\pm 0.15}$ | $87.14_{\pm 0.12}$ |
| 50 | $69.14_{\pm 0.15}$ | $86.12_{\pm 0.15}$ | $72.25_{\pm 0.15}$ | $88.32_{\pm 0.12}$ |
| 100 | $69.13_{\pm 0.15}$ | $86.07_{\pm 0.10}$ | $72.21_{\pm 0.15}$ | $88.37_{\pm 0.12}$ |
| 1000 | $69.11_{\pm 0.15}$ | $86.11_{\pm 0.10}$ | $72.12_{\pm 0.15}$ | $88.13_{\pm 0.12}$ |

**Visualization of the prototype adaptation process.** We undertake a qualitative analysis of the adaptation process of our ProtoDiff framework during the meta-test time. We employ a 3-way 5-shot setup on the *mini*ImageNet dataset for more comprehensive visualization of the adaptation process. To achieve this, we depict the vanilla prototypes, the diffused prototypes at various timesteps, and instances from the support and query sets. All examples are first normalized to a length of 1 and

**Table 5: Cross-domain few-shot** comparison on four benchmarks. ProtoDiff is a consistent top-performer.

| Method | CropDiseases | | EuroSAT | | ISIC | | ChestX | |
|---|---|---|---|---|---|---|---|---|
| | 1-shot | 5-shot | 1-shot | 5-shot | 1-shot | 5-shot | 1-shot | 5-shot |
| ProtoNet [42] | $57.57_{\pm0.51}$ | $79.72_{\pm0.67}$ | $54.19_{\pm0.57}$ | $73.29_{\pm0.71}$ | $29.62_{\pm0.32}$ | $39.57_{\pm0.57}$ | $22.30_{\pm0.25}$ | $24.05_{\pm1.01}$ |
| GNN [40] | $57.19_{\pm0.50}$ | $83.12_{\pm0.40}$ | $54.61_{\pm0.50}$ | $78.69_{\pm0.40}$ | $30.14_{\pm0.30}$ | $42.54_{\pm0.40}$ | $21.94_{\pm0.20}$ | $23.87_{\pm0.20}$ |
| AFA [20] | $67.61_{\pm0.50}$ | $88.06_{\pm0.30}$ | $63.12_{\pm0.50}$ | $85.58_{\pm0.40}$ | $33.21_{\pm0.30}$ | $\mathbf{46.01}_{\pm0.40}$ | $22.92_{\pm0.20}$ | $25.02_{\pm0.20}$ |
| ATA [51] | $67.47_{\pm0.50}$ | $\mathbf{90.59}_{\pm0.30}$ | $61.35_{\pm0.50}$ | $83.75_{\pm0.40}$ | $33.21_{\pm0.30}$ | $44.91_{\pm0.40}$ | $22.10_{\pm0.20}$ | $24.32_{\pm0.20}$ |
| HVM [11] | $65.13_{\pm0.45}$ | $87.65_{\pm0.31}$ | $61.97_{\pm0.34}$ | $74.88_{\pm0.45}$ | $33.87_{\pm0.35}$ | $42.05_{\pm0.34}$ | $\mathbf{22.94}_{\pm0.47}$ | $27.15_{\pm0.45}$ |
| *This paper* | $\mathbf{68.93}_{\pm0.31}$ | $90.15_{\pm0.31}$ | $\mathbf{65.93}_{\pm0.34}$ | $\mathbf{87.25}_{\pm0.35}$ | $\mathbf{34.97}_{\pm0.33}$ | $45.65_{\pm0.31}$ | $23.01_{\pm0.45}$ | $\mathbf{28.54}_{\pm0.41}$ |

subsequently projected onto a two-dimensional space using t-SNE [48]. Figure 7 displays the instance representations from both the query and support sets that are cross and circle, the vanilla prototypes that are denoted by a four-point star symbol, an the diffused prototypes that are denoted by stars. Arrows depict the adaptation process of the diffused prototypes, and each color corresponds to each respective class. Initially, the diffused prototypes are randomly allocated based on a Gaussian distribution, positioning them considerably from the optimal prototypes for each class. As the diffusion process contin-

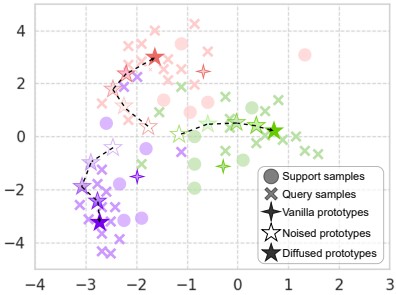

**Figure 7: t-SNE visualization of ProtoDiff adaptation process** on 3-way 5-shot task.

ues, these diffused prototypes incrementally approach the optimal prototypes, resulting in more distinctive representations for different classes. In contrast, the vanilla prototypes lack distinguishability as they are computed using a single deterministic function based solely on the support set. This observation underscores the importance of the prototype adaptation process in attaining improved performance in few-shot classification tasks.

**Within-domain few-shot.** We evaluate our method on few-shot classification within domains, in which the training domain is consistent with the test domain. The results are reported in Table 4. In this comparison, we apply Set-Feat [1] with ProtoDiff to experiment since SetFeat is the current state-of-the-art model based on prototype-based meta-learning. Our ProtoDiff consistently achieves state-of-the-art performance on all datasets under various shots. The better performance confirms that ProtoDiff can achieve more informative and higher quality prototypes to perform better for few-shot learning within domains.

**Table 4: Within-domain few-shot** comparison on *mini*Imagenet and *tiered*Imagenet. ProtoDiff performs well on all datasets.

| Method | *mini*Imagenet | | *tiered*Imagenet | |
|---|---|---|---|---|
| | 1-shot | 5-shot | 1-shot | 5-shot |
| TapNet [59] | $61.65_{\pm0.15}$ | $76.36_{\pm0.10}$ | $63.08_{\pm0.15}$ | $80.26_{\pm0.12}$ |
| CTM [26] | $62.05_{\pm0.55}$ | $78.63_{\pm0.06}$ | $64.78_{\pm0.11}$ | $81.05_{\pm0.52}$ |
| MetaOptNet [24] | $62.64_{\pm0.61}$ | $78.63_{\pm0.46}$ | $65.81_{\pm0.74}$ | $81.75_{\pm0.53}$ |
| Meta-Baseline [5] | $63.17_{\pm0.23}$ | $79.26_{\pm0.17}$ | $68.62_{\pm0.27}$ | $83.74_{\pm0.18}$ |
| CAN [18] | $63.85_{\pm0.48}$ | $79.44_{\pm0.34}$ | $69.89_{\pm0.51}$ | $84.23_{\pm0.37}$ |
| Meta DeepBDC [53] | $67.34_{\pm0.43}$ | $84.46_{\pm0.28}$ | $72.34_{\pm0.49}$ | $87.31_{\pm0.32}$ |
| SUN [9] | $67.80_{\pm0.45}$ | $83.25_{\pm0.30}$ | $72.99_{\pm0.50}$ | $86.74_{\pm0.33}$ |
| SetFeat [1] | $68.32_{\pm0.62}$ | $82.71_{\pm0.46}$ | $73.63_{\pm0.88}$ | $87.59_{\pm0.57}$ |
| *This paper* | $\mathbf{71.25}_{\pm0.45}$ | $\mathbf{83.95}_{\pm0.45}$ | $\mathbf{75.97}_{\pm0.75}$ | $\mathbf{88.75}_{\pm0.18}$ |

**Cross-domain few-shot.** We also evaluate our method in the cross-domain few-shot classification, where the training domain is different from test domain. Table 5 shows the evaluation of ProtoDiff on four datasets, with 5-way 1-shot and 5-way 5-shot settings. ProtoDiff also achieves competitive performance on all four cross-domain few-shot learning benchmarks for each setting. On EuroSAT [15], our model obtains high recognition accuracy under different shot configurations, outperforming the second best method, AFA [20], by a significant margin of $2.82\%$ for 5-way 1-shot. Even on the most challenging ChestX [52], which has a considerable domain gap with *mini*ImageNet, our model achieves $28.54\%$ accuracy in the 5-way 5-shot setting, surpassing the second-best HVM [11] by $1.39\%$. The consistent improvement across all benchmarks and settings confirms the effectiveness of ProtoDiff for cross-domain few-shot learning.

**Few-task few-shot.** We evaluate ProtoDiff on the four different datasets under 5-way 1-shot and 5-way 5-shot in Table 6. In this comparison, we apply MLTI [56] with ProtoDiff to experiment. Our method achieves state-of-the-art performance on all four few-task meta-learning benchmarks under each setting. On *mini*Imagenet-S, our model achieves $44.15\%$ under 1-shot, surpassing the

**Table 6: Few-task few-shot** comparison on four benchmarks. ProtoDiff consistently ranks among the top-performing models.

| | *mini*Imagenet-S | | ISIC | | Dermnet-S | | Tabular Murris | |
|---|---|---|---|---|---|---|---|---|
| | **1-shot** | **5-shot** | **1-shot** | **5-shot** | **1-shot** | **5-shot** | **1-shot** | **5-shot** |
| ProtoNet [42] | $36.26_{\pm0.70}$ | $50.72_{\pm0.70}$ | $58.56_{\pm1.01}$ | $66.25_{\pm0.96}$ | $44.21_{\pm0.75}$ | $60.33_{\pm0.70}$ | $80.03_{\pm0.90}$ | $89.20_{\pm0.56}$ |
| Meta-Dropout [23] | $38.32_{\pm0.75}$ | $52.53_{\pm0.70}$ | $58.40_{\pm1.16}$ | $67.32_{\pm0.95}$ | $44.30_{\pm0.81}$ | $60.86_{\pm0.75}$ | $78.18_{\pm0.90}$ | $89.25_{\pm0.56}$ |
| MetaMix [55] | $39.80_{\pm0.87}$ | $53.35_{\pm0.88}$ | $59.66_{\pm1.10}$ | $68.97_{\pm0.80}$ | $46.06_{\pm0.85}$ | $62.97_{\pm0.70}$ | $79.56_{\pm0.91}$ | $88.88_{\pm0.60}$ |
| Meta Interpolation [25] | $40.28_{\pm0.70}$ | $53.06_{\pm0.72}$ | - | - | - | - | - | - |
| MLTI [56] | $41.36_{\pm0.73}$ | $55.34_{\pm0.72}$ | $62.82_{\pm1.09}$ | $71.52_{\pm0.89}$ | $49.38_{\pm0.85}$ | $65.19_{\pm0.73}$ | $81.89_{\pm0.88}$ | $90.12_{\pm0.59}$ |
| MetaModulation [45] | $43.21_{\pm0.73}$ | $57.26_{\pm0.72}$ | $65.61_{\pm1.09}$ | $\mathbf{76.40}_{\pm0.89}$ | $50.45_{\pm0.84}$ | $67.05_{\pm0.73}$ | $83.13_{\pm0.89}$ | $91.23_{\pm0.57}$ |
| *This paper* | $\mathbf{44.75}_{\pm0.70}$ | $\mathbf{58.18}_{\pm0.72}$ | $\mathbf{66.13}_{\pm1.04}$ | $76.23_{\pm0.81}$ | $\mathbf{51.53}_{\pm0.85}$ | $\mathbf{67.97}_{\pm0.74}$ | $\mathbf{84.03}_{\pm0.89}$ | $\mathbf{91.35}_{\pm0.57}$ |

best method MetaModulation [45], by a margin of 1.54%. Even on the most challenging DermNet-S, which forms the largest dermatology dataset, our model delivers 51.53% on the 5-way 1-shot setting. The consistent improvements on all benchmarks under various configurations confirm that our approach is also effective for few-task meta-learning.

**Limitations.** Naturally, our proposal also comes with limitations. Firstly, the diffusion model employed in our approach necessitates a substantial number of timesteps to sample the diffused prototype during the meta-test stage. Although we somewhat alleviate this issue by utilizing DDIM [44] sampling, it still requires more computational resources than vanilla prototypes. Secondly, obtaining the overfitted prototype involves fine-tuning, which inevitably leads to increased training time. While this fine-tuning step contributes to the effectiveness of our model, it comes with an additional computational cost. In meta-training and meta-testing ProtoDiff is slower by factors of ProtoNet $5\times$ and $15\times$ in terms of wall-clock times per task. More detailed time numbers are provided in the Appendix. As part of future work, we will investigate and address these limitations to further enhance the applicability and efficiency of our approach.

## 6 Conclusion

We proposed ProtoDiff, a novel approach to prototype-based meta-learning that utilizes a task-guided diffusion model during the meta-training phase to generate efficient class representations. We addressed the limitations of estimating a deterministic prototype from a limited number of examples by optimizing a set of prototypes to accurately represent individual tasks and training a task-guided diffusion process to model each task's underlying distribution. Our approach considers both probabilistic and task-guided prototypes, enabling efficient adaptation to new tasks while maintaining the informativeness and scalability of prototypical networks. Extensive experiments on three distinct few-shot learning scenarios: within-domain, cross-domain, and few-task few-shot learning validated the effectiveness of ProtoDiff. Our results demonstrated significant improvements in classification accuracy compared to state-of-the-art meta-learning techniques, highlighting the potential of task-guided diffusion in augmenting few-shot learning and advancing meta-learning.

## Acknowledgment

This work is financially supported by the Inception Institute of Artificial Intelligence, the University of Amsterdam and the allowance Top consortia for Knowledge and Innovation (TKIs) from the Netherlands Ministry of Economic Affairs and Climate Policy.

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

# A Algorithms

We describe the detailed algorith ms for meta-training and meta-test of ProtoDiff as following Algorithm 1 and 2, respectively:

---

**Algorithm 1** Meta-training phase of ProtoDiff.

**Input:** $p(\mathcal{T})$: distribution over tasks; $\theta$: diffusion parameters; $\phi$: feature extractor parameters; $T$: diffusion steps.

**Output:** $\mathbf{z}_\theta$: diffusion network, $f_\phi$: feature extractor.

---

1:  **repeat**
2:      Sample batch of tasks $\mathcal{T}^i \sim p(\mathcal{T})$
3:      **for** all $\mathcal{T}^i$ **do**
4:          Sample support and query set $\{\mathcal{S}^i, \mathcal{Q}^i\}$ from $\mathcal{T}^i$
5:          Compute the vanilla prototype $\tilde{\mathbf{z}}^i = f(\mathcal{S}^i)$
6:          **repeat**
7:              Take some gradient step on $\nabla\mathcal{L}_{\mathrm{CE}}(\mathcal{S}^i, \mathcal{Q}^i)$, updating $\phi^i$
8:          **until** $\mathcal{L}_{\mathrm{CE}}(\mathcal{S}_i, \mathcal{Q}_i)$ converges
9:          Compute the overfitted prototype by using the updated $\phi^i$, $\mathbf{z}^{i,*} = f_{\phi^i}(\mathcal{S}_i)$
10:         $t \sim Uniform(1...T)$
11:         $\epsilon = \mathcal{N}(\mathbf{0}, \mathbf{1})$
12:         $\beta_t = \frac{10^{-4}(T-t)+10^{-2}(t-1)}{T-1}, \alpha_t = 1 - \beta_t, \bar{\alpha}_t = \Pi_{k=0}^{k=t}\alpha_k$
13:         $\hat{\mathbf{z}}_t^i = \sqrt{\bar{\alpha}_t}(\mathbf{z}^{i,*} - \tilde{\mathbf{z}}^i) + \sqrt{1 - \bar{\alpha}_t}^2\epsilon$
14:         Compute diffusion loss $\mathcal{L}_{\mathrm{diff}}$ with Equation (9) and the final loss $\mathcal{L}_{\mathcal{T}^i}$ with Equation (10)
15:         Update $\{\phi, \theta\} \leftarrow \{\phi, \theta\} - \beta\nabla_{\{\phi,\theta\}}\sum_{\mathcal{T}^i \sim p(\mathcal{T})}\mathcal{L}_{\mathcal{T}^i}$ using query data of each task.
16:     **end for**
17: **until** $f_\phi$ and $\mathbf{z}_\theta$ converges

---

**Algorithm 2** Meta-test phase of ProtoDiff.

**Input:** $\tau = \{\mathcal{S}, \mathcal{Q}\}$: meta-test task, $\mathbf{z}_\theta$: trained diffusion network parameters, $f_\phi$: trained feature extractor network parameters, $T$: diffusion steps.

---

1: $\mathbf{z}_T \sim \mathcal{N}(\mathbf{0}, \mathbf{I})$
2: Compute vanilla prototype $\tilde{\mathbf{z}}$ with support set $f(\mathcal{S})$
3: **for** t=$T, \cdots, 1$ **do**
4:     $\epsilon = N(\mathbf{0}, \mathbf{1})$
5:     $\beta_t = \frac{10^{-4}(T-t)+10^{-2}(t-1)}{T-1}, \alpha_t = 1 - \beta_t, \bar{\alpha}_t = \Pi_{k=0}^{k=t}\alpha_k$
6:     $\mathbf{z}_{t-1} = \mathbf{z}_\theta(\mathbf{z}_t, \tilde{\mathbf{z}}, t)$
7: **end for**
8: Compute the final prediction $\mathbf{y}^q$ by with Equation (1) based on $\mathbf{z}_0 + \tilde{\mathbf{z}}$

---

## B Per-task prototype overfitting architecture

To enhance our comprehension of the Per-task prototype overfitting part, we propose a succinct architectural representation depicted in Figure 8. The initial step entails the computation of the conventional prototypes $\tilde{\mathbf{z}}$ for a meta-training task. Subsequently, Equation (1) is employed to calculate the predictions for the query sample. The backbone's parameters are subsequently updated through $I$ iterations. Through the utilization of parameters from the final iteration, we ultimately obtain the prototypes $\mathbf{z}^*$ that exhibit overfitting characteristics.

## C Residual prototype learning architecture

In order to gain a more comprehensive understanding of our residual prototype learning, we have crafted a succinct architecture diagram illustrated in Figure 9. Our proposition involves the prediction of the prototype update, denoted as $\mathbf{z}^* - \tilde{\mathbf{z}}$, instead of directly predicting the overfitted prototype

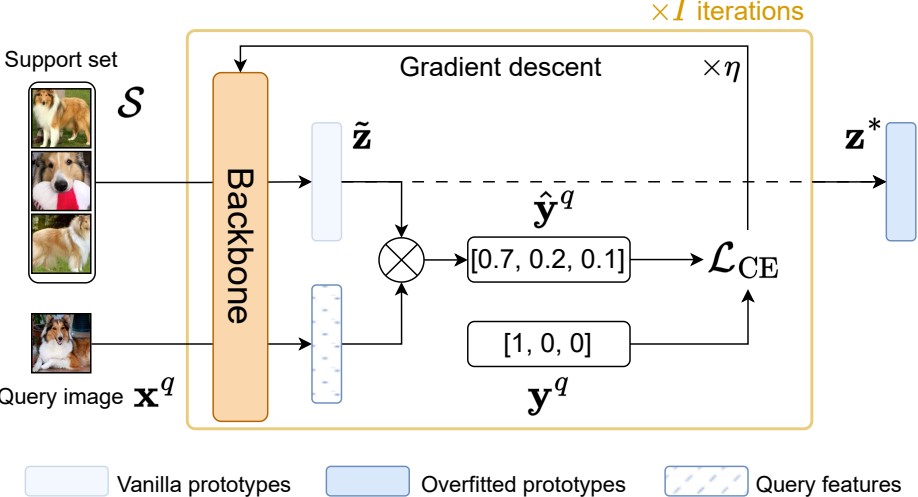

Figure 8: Per-task prototype overfitting.

$\mathbf{z}^*$. This distinctive approach also allows us to initialize ProtoDiff with the capability to perform the identity function, achieved by assigning zero weights to the decoder. Notably, we have discovered that the amalgamation of a global residual connection and the identity initialization substantially expedites the training process. By harnessing this mechanism, we have successfully enhanced the performance of ProtoDiff in the context of few-shot learning tasks.

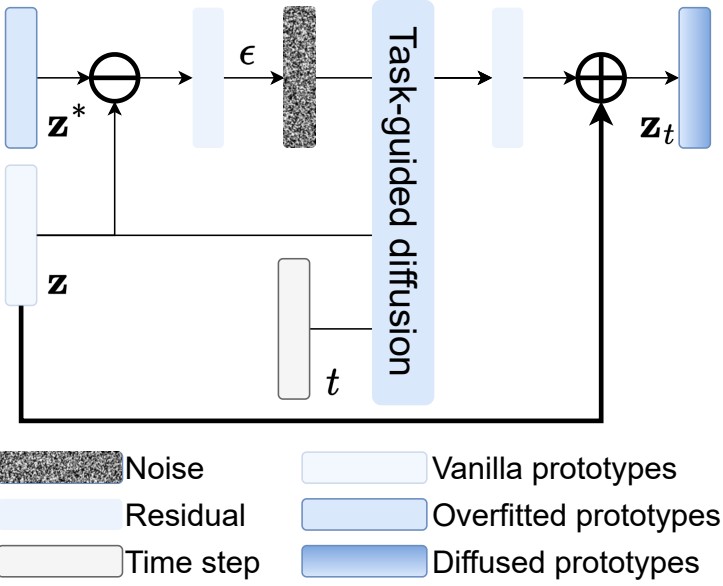

Figure 9: Residual prototype learning.

## D  Datasets

**Within-domain few-shot.** For this setting we focus on 5-way 1-shot/5-shot tasks, which aligns with previous research [42]. The within-domain few-shot experiments are performed on three datasets: *mini*Imagenet [50] *tiered*Imagenet [35], and ImageNet-800 [5]. *mini*Imagenet consists of 100 randomly selected classes from ILSVRC-2012 [37], while *tiered*Imagenet is composed of 608 classes that are grouped into 34 high-level categories. We measure the accuracy of 600 tasks randomly

sampled from the meta-test set to evaluate the performance. Following [5], we also evaluate our model on ImageNet-800, a dataset obtained by randomly dividing the 1,000 classes of ILSVRC-2012 into 800 base classes and 200 novel classes. The base classes consist of images from the original training set, while the novel classes comprise images from the original validation set.

**Cross-domain few-shot.** In the 5-way 5-shot cross-domain few-shot classification experiments, the training domain is *mini*Imagenet [50], and the testing is conducted on four different domains. These domains are CropDisease [30], which contains plant disease images; EuroSAT [15], a collection of satellite images; ISIC2018 [47], consisting of dermoscopic images of skin lesions, and ChestX [52], a dataset of X-ray images.

**Few-task few-shot.** Few-task few-shot learning [56] challenges the common meta-training assumption of having many tasks available. We conductt experiments on four few-task meta-learning challenges, namely *mini*Imagenet-S [50], ISIC [29], DermNet-S [6], and Tabular Murris [3]. To reduce the number of tasks and make it comparable to previous works, we followed [56] by limiting the number of meta-training classes to 12 for miniImagenet-S, with 20 meta-test classes. ISIC [29] involves classifying dermoscopic images across nine diagnostic categories, with 10,015 images available for training in eight different categories, of which we selected four as meta-training classes. DermNet [6], which contains over 23,000 images of skin diseases, was utilized to construct Dermnet-S by selecting 30 diseases as meta-training classes, following [56]. Finally, Tabular Murris [3], which deals with cell type classification across organs and includes nearly 100,000 cells from 20 organs and tissues, was utilized to select 57 base classes as meta-training classes, again following the same approach as [56].

## E    Implementation details

In our within-domain experiments, we utilize a Conv-4 and ResNet-12 backbone for *mini*Imagenet and *tiered*Imagenet. A ResNet-50 is used for ImageNet-800. We follow the approach described in [5] to achieve better performance and initially train a feature extractor on all the meta-training data without episodic training. We use the SGD optimizer with a momentum of 0.9, a learning rate starting from 0.1, and a decay factor of 0.1. For *mini*Imagenet, we train for 100 epochs with a batch size of 128, where the learning rate decays at epoch 90. For tieredImageNet, we train for 120 epochs with a batch size of 512, where the learning rate decays at epochs 40 and 80. Lastly, for ImageNet-800, we train for 90 epochs with a batch size of 256, where the learning rate decays at epochs 30 and 60. The weight decay is 0.0005 for ResNet-12 and 0.0001 for ResNet-50. Standard data augmentation techniques, including random resized crop and horizontal flip, are applied. For episodic training, we use the SGD optimizer with a momentum of 0.9, a fixed learning rate of 0.001, and a batch size of 4, meaning each training batch consists of 4 few-shot tasks to calculate the average loss. For our cross-domain experiments, we use a ResNet-10 backbone to extract image features, which is a common choice for cross-domain few-shot classification [14, 57]. The training configuration for this experiment is the same as the within-domain *mini*Imagenet training. For few-task few-shot learning, we follow [56] using a network containing four convolutional blocks and a classifier layer. Each block comprises a 32-filter $3 \times 3$ convolution, a batch normalization layer, a ReLU nonlinearity, and a $2 \times 2$ max pooling layer. All experiments are performed on a single A100 GPU, each taking approximately 20 hours. We will release all our code.

## F    Visualization of diffusion process

The ProtoDiff method utilizes a task-guided diffusion model to generate prototypes that provide efficient class representations, as discussed in the previous section. To better understand the effectiveness of our proposed approach, we provide a visualization by Grad-Cam [41] in Figure of the diffusion process, demonstrating how ProtoDiff gradually aggregates towards the desired class prototype during meta-training. The vanilla prototype is shown in the first row on the left, which does not exclusively focus on the *guitar*. In contrast, the overfitted prototype in the second row on the left provides the highest probability for the *guitar*. ProtoDiff, with the diffusion process, randomly selects certain areas to add noise and perform diffusion, resulting in a prototype that gradually moves towards the *guitar* with the highest probability at t=0. Moreover, ProtoDiff with residual learning produces a more precise prototype. The comparison between these different prototypes demonstrates

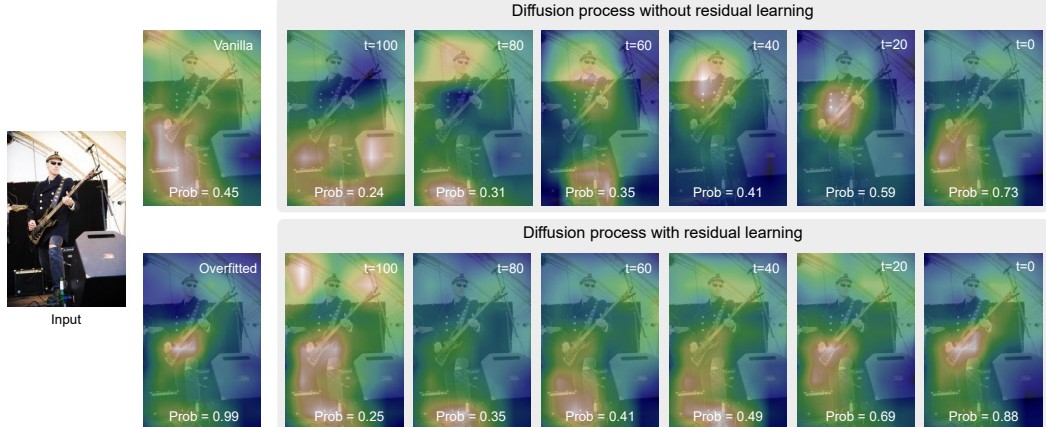

**Figure 10: Visualization of the diffusion process.** The first row on the right shows the vanilla prototype, which does not exclusively focus on the *guitar*. In contrast, the overfitted prototype in the second row on the right provides the highest probability for the *guitar*. ProtoDiff randomly selects certain areas to predict during the diffusion process, with the lowest probability at the beginning time step. As time progresses, the prototype gradually aggregates towards the *guitar*, with the highest probability at t=0. In comparison, ProtoDiff with residual learning produces a more precise prototype.

the effectiveness of the ProtoDiff diffusion process in generating a more accurate and informative prototype for few-shot learning tasks.

## G    More results

**Effect of transformer structure** Our ProtoDiff model is constructed using a design inspired by GPT-2. This includes a 12-layer transformer, a 512-dimensional linear transformation, an attention mechanism with 16 heads, and an MLP with a hidden dimensionality 512. These configurations are in line with GPT-2's default parameters. The configuration files can be accessed in our code repository for more detailed parameter setup. Our experiments in the table 7a and 7b highlight that our model achieves optimal performance with these settings.

**Table 7:** Effect of transformer structures

**(a)** Results on the different transformer structures.

| Structures | *mini*Imagenet | |
| --- | --- | --- |
| | 1-shot | 5-shot |
| 3-layers | $62.17_{\pm 0.25}$ | $78.93_{\pm 0.17}$ |
| 6-layers | $63.25_{\pm 0.22}$ | $79.63_{\pm 0.15}$ |
| 9-layers | $65.21_{\pm 0.21}$ | $80.13_{\pm 0.18}$ |
| 12-layers | $66.63_{\pm 0.21}$ | $83.48_{\pm 0.15}$ |
| 15-layers | $\mathbf{64.15}_{\pm 0.25}$ | $\mathbf{80.93}_{\pm 0.18}$ |

**(b)** Results on the different heads numbers.

| Nodes numbers | *mini*Imagenet | |
| --- | --- | --- |
| | 1-shot | 5-shot |
| node = 1 | $63.28_{\pm 0.24}$ | $79.97_{\pm 0.15}$ |
| node = 4 | $64.43_{\pm 0.22}$ | $80.13_{\pm 0.14}$ |
| node = 8 | $65.91_{\pm 0.23}$ | $81.91_{\pm 0.16}$ |
| node = 16 | $\mathbf{66.63}_{\pm 0.21}$ | $\mathbf{83.48}_{\pm 0.15}$ |

**Effect of different time steps** We have selected the diffusion time step T for the diffusion process to be 100. We've adopted the DDIM sampling strategy to accelerate prediction with . This effectively reduces the total sample requirement to just 10. We've conducted comparative experiments using varying diffusion times and intermediate intervals. As presented in the tables 8 and 9, we observe that as the diffusion timesteps increase, both the performance and the inference time increase. Simultaneously, when is increased, the inference time decreases, making the process more efficient.

**Effect of more support images** To prove more support images during meta-training to obtain more accurate prototypes, we conducted an experimental comparison using different numbers of support sets during meta-training. The results in the table 10 illustrate that augmenting the number of support images for each class during the meta-training phase enhances performance across various shots.

**Table 8:** Results with the different diffusion timesteps on *mini*Imagenets.

| dim($\tau$) = 10, Timesteps | 1-shot | 5-shot | 1-shot inference time | 5-shot inference time |
|---|---|---|---|---|
| T =10 | 63.98$\pm$0.24 | 80.12 $\pm$0.15 | 1 ms | 2 ms |
| T = 50 | 65.93$\pm$0.21 | 82.93 $\pm$0.17 | 5 ms | 7 ms |
| T = 100 | 66.63$\pm$0.21 | 83.48$\pm$0.15 | 9 ms | 14 ms |
| T = 500 | 66.65$\pm$0.23 | 83.59$\pm$0.14 | 53 ms | 93 ms |
| T = 1000 | 66.74$\pm$0.20 | 83.97$\pm$0.17 | 102 ms | 192 ms |

**Table 9:** Results with the different diffusion timesteps on *mini*Imagenets.

| dim($\tau$), T = 100 | 1-shot | 5-shot | Speed |
|---|---|---|---|
| $\tau$ =1 | 66.78$\pm$0.21 | 83.62 $\pm$0.14 | 99 ms |
| $\tau$ = 5 | 66.15$\pm$0.23 | 83.44$\pm$0.16 | 45 ms |
| $\tau$ = 10 | 66.63$\pm$0.21 | 83.48 $\pm$0.15 | 9 ms |
| $\tau$ = 20 | 66.12$\pm$0.21 | 82.79 $\pm$0.13 | 5 ms |
| $\tau$ = 100 | 63.15$\pm$0.23 | 81.27 $\pm$0.14 | 1 ms |

**Meta-training and meta-test wall-clocks times per task** We have compiled the wall-clock time for both the meta-training and meta-testing phases of ProtoDiff and compared these against the respective times for ProtoNet in the Tables 11a and 11b. In meta-training and meta-testing wall-clock times per task, our ProtoDiff is slower by factors of ProtoNet 5$\times$ and 15 $\times$, respectively. As part of future work, we will investigate and address this limitation to further enhance the efficiency of our approach.

**Rationale for Conditioning** We utilize the "vanilla" prototype as a guiding condition, gradually enabling the diffusion model to generate class-specific prototypes. We also explored two alternative strategies for conditioning in the table 12: one without any conditions and another with learned class embeddings. When not conditioned, the performance tends to be arbitrary due to the absence of class-specific cues during diffusion. On the other hand, employing learned class embeddings as a condition yielded subpar results compared to the vanilla prototype, potentially due to outliers in each class's learned embeddings.

**Results of our pre-trained model** We also give the results of our own pre-trained model before applying ProtoDiff to provide a clear comparison and demonstrate our method's improvements. We have prepared the results of our pre-trained model (trained with CE loss on the whole training set) and will present it in table 13.

**Table 10:** Results of the different support sets for each class.

| | *mini*Imagenet | |
|---|---|---|
| **Structures** | **1-shot** | **5-shot** |
| 1 | $66.63_{\pm 0.21}$ | $73.12_{\pm 0.15}$ |
| 5 | $70.25_{\pm 0.22}$ | $83.48_{\pm 0.13}$ |
| 8 | $73.91_{\pm 0.21}$ | $84.72_{\pm 0.13}$ |
| 16 | $82.21_{\pm 0.23}$ | $84.98_{\pm 0.17}$ |

**Table 11:** Meta-training and meta-test wall-clock times on each task.

**(a)** Meta-training.

| | *mini*Imagenet | |
|---|---|---|
| **Meta-training** | **1-shot** | **5-shot** |
| ProtoNet | 0.6 ms | 1.2 ms |
| ProtoDiff | 3 ms | 5.3 ms |

**(b)** Meta-test.

| | *mini*Imagenet | |
|---|---|---|
| **Meta-test** | **1-shot** | **5-shot** |
| ProtoNet | 0.6 ms | 1.2 ms |
| ProtoDiff | 9 ms | 14 ms |

**Table 12:** Results of different conditions.

| | *mini*Imagenet | |
|---|---|---|
| **Conditions** | **1-shot** | **5-shot** |
| w/o conditions | $20.27_{\pm 0.25}$ | $22.32_{\pm 0.15}$ |
| Learnt class embeddings | $66.17_{\pm 0.22}$ | $81.17_{\pm 0.16}$ |
| Vanilla prototype | $66.63_{\pm 0.21}$ | $83.48_{\pm 0.15}$ |

**Table 13:** Results of our pre-trained model on different datasets

| | | *mini*Imagenet | | *tiered*Imagenet | | Imagenet-800 | |
|---|---|---|---|---|---|---|---|
| | | **1-shot** | **5-shot** | **1-shot** | **5-shot** | **1-shot** | **5-shot** |
| Pretrained from Chen *et al.* [5] | Classifier-Baseline | $58.91_{\pm 0.23}$ | $77.76_{\pm 0.17}$ | $68.07_{\pm 0.29}$ | $83.74_{\pm 0.18}$ | $86.07_{\pm 0.21}$ | $96.14_{\pm 0.08}$ |
| Pretrained from Chen *et al.* [5] | **ProtoDiff** | $\mathbf{66.63}_{\pm 0.21}$ | $\mathbf{83.48}_{\pm 0.15}$ | $\mathbf{72.95}_{\pm 0.24}$ | $\mathbf{85.15}_{\pm 0.18}$ | $\mathbf{92.13}_{\pm 0.20}$ | $\mathbf{98.21}_{\pm 0.08}$ |
| Pretrained from ours | Classifier-Baseline | $58.75_{\pm 0.21}$ | $77.86_{\pm 0.17}$ | $68.15_{\pm 0.28}$ | $83.95_{\pm 0.17}$ | $85.97_{\pm 0.22}$ | $96.03_{\pm 0.09}$ |
| Pretrained from ours | **ProtoDiff** | $\mathbf{66.49}_{\pm 0.25}$ | $\mathbf{83.51}_{\pm 0.14}$ | $\mathbf{73.01}_{\pm 0.24}$ | $\mathbf{85.72}_{\pm 0.17}$ | $\mathbf{92.05}_{\pm 0.21}$ | $\mathbf{98.11}_{\pm 0.18}$ |

