# OpenReview forum: "ProtoDiff: Learning to Learn Prototypical Networks by Task-Guided Diffusion"
_NeurIPS.cc/2023/Conference — NeurIPS 2023 poster_

### Official Review · Reviewer_BNd7 · 2023-06-28

**Soundness:** 3 good
**Presentation:** 3 good
**Contribution:** 2 fair
**Rating:** 7
**Confidence:** 4

**Summary:**

The paper introduces ProtoDiff, a novel addition to the family of Prototypical Network methods that uses a diffusion model to obtain better class prototypes at inference time.

Prototypical Networks perform classification with metric-based comparisons between examples and “class prototypes”, variables computed based on support sets of class examples.

In ProtoDiff, a “vanilla” prototype is first computed based on the class support set. This vanilla prototype is then used as conditional input of a diffusion model, computing over T steps a residual between the vanilla prototype and a better, “diffused” prototype for the class. This diffused prototype is then used to perform traditional prototypical network classification. To train the diffusion model at meta-training time, one has to produce “overfitted” prototypes for tasks, based on both the query set and support set for the task.

The authors then demonstrate the effectiveness of their approach with a series of experiments for within-domain few-shot learning, cross-domain few-shot learning, and few-task few-shot learning.

**Strengths:**

The combination of prototypical networks and diffusion models is interesting: applying insights from recent developments in diffusion models to the field of meta-learning is still under-explored in the field.

The paper is clearly presented, the procedure itself is a novel combination of existing ideas, and is clearly motivated and easily argued for.

The experiments seem to support the claim that the diffusion model specifically is what is needed to edge out any remaining performance missing from the original prototypical network approach, in the form of a residual.

**Weaknesses:**

Overall, the paper doesn’t have any glaring flaw, if not for the fact that experimental gains seems marginal. In practice, the idea behind the paper seems to be about squeezing any additional percentage point of performance out of the existing technique of prototypical networks. This is especially apparent with the fact that the diffusion model is used to produce a residual, and is also itself conditioned on the “vanilla” prototype.

As for related work, it would be interesting to explore the connection with *Nava et al. (2022). Meta-Learning via Classifier (-free) Diffusion Guidance.* (https://arxiv.org/abs/2210.08942). They also apply diffusion models in the context of meta-learning, using a similar metric classification technique for the downstream tasks, even though they approach the problem with the goal of conditionally diffusing a model’s parameters, instead of class prototypes.

**Questions:**

I would ask the authors to further elaborate on the positioning of their work within the meta-learning literature, arguing for the choice of conditioning the diffusion model on the existing “vanilla” prototype, and comparing it with possible alternative choices, such as specifically learnt class embeddings.

**Limitations:**

The authors satisfactorily discuss the limitations of their work in the main paper text.

---

> ### Author Rebuttal · Authors · 2023-08-09
>
> ***Thank you for your time, effort, and constructive comments.***
>
> * **Weakness**
>
> Answer:  Thank you for the thoughtful feedback and the reference to the work by Nava et al. We acknowledge the concern that the experimental gains might seem marginal and that the focus appears on incremental improvements within prototypical networks.
>
> However, it is essential to highlight that applying a diffusion model in ProtoDiff represents a novel way to generate prototypes, moving beyond simple performance squeezing. By using a task-guided diffusion model to produce a residual and allowing for task-specific prototype generation, ProtoDiff opens a new avenue in prototype learning. This approach offers a subtle but significant departure from conventional methods, allowing for more robust and nuanced class representations.
>
> As for the connection with the paper by Nava et al., we agree that exploring similarities and differences between our approach and theirs would enrich our work. While our focus is on the diffusion of class prototypes, their method applies diffusion models in a broader context of the model's parameters. We will review this paper in depth and include a discussion in our revised manuscript to clarify how ProtoDiff fits within this broader research landscape.
>
> * **Questions**
>
> Answer:
> **Position in the Meta-Learning Landscape**: As far as we know, our ProtoDiff is the first approach that integrates a diffusion model for generating prototypes in few-shot meta-learning. This design choice facilitates seamless integration with any metric-based meta-learning method for mutual benefit.
>
>
> **Rationale for Conditioning**: We utilize the “vanilla” prototype as a guiding condition, gradually enabling the diffusion model to generate class-specific prototypes. We also explored two alternative strategies for conditioning in the following table: one without any conditions and another with learned class embeddings.
> When not conditioned, the performance tends to be arbitrary due to the absence of class-specific cues during diffusion. On the other hand, employing learned class embeddings as a condition yielded subpar results compared to the vanilla prototype, potentially due to outliers in each class's learned embeddings.
> We plan to elaborate on these insights in our revised manuscript.
>
>
>
> |Conditions|1-shot|5-shot|
> |:-----:|:-----:|:-----:|
> |W/o conditons|20.27$\pm$ 0.25|22.32$\pm$0.15|
> |Learnt class embeddings|66.17$\pm$ 0.22|81.17$\pm$0.16|
> |Vanilla prototype|66.63$\pm$ 0.21|83.48$\pm$0.15|

---

> > ### Comment · Reviewer_BNd7 · 2023-08-21
> > **Reply to the Authors**
> >
> > We thank the Authors for their detailed response, which addresses the concerns I raise. I look forward to a broader comparison with the research landscape in the final version.
> > Meanwhile, I am willing to raise my score to a 7.

---

> > > ### Author Response · Authors · 2023-08-21
> > >
> > > We appreciate your feedback and are pleased to hear that our detailed response has addressed your concerns. We will certainly incorporate a broader comparison with the research landscape in the final version of the manuscript. Your willingness to raise the score to a 7 is greatly appreciated.

---

### Official Review · Reviewer_8tyC · 2023-07-04

**Soundness:** 3 good
**Presentation:** 3 good
**Contribution:** 4 excellent
**Rating:** 7
**Confidence:** 3

**Summary:**

This paper proposes using the diffusion model to learn a prototypical network. During the meta-train stage, an overfitted prototype (by fine-tuning the classifier until it gets a probability of 1 for the task) is obtained first, then given a vanilla prototype (by simply averaging the features) as input, the diffusion model is to denoise a random prototype into the overfitted prototype. This is because the overfitted prototype is unknown during the meta-test stage, therefore the diffusion model is to obtain this overfitted prototype by using the vanilla prototype as conditional information.

**Strengths:**

- the whole design of leveraging the diffusion model to predict the prototype is quite interesting and novel. That is, overfitted prototype indeed is better than the vanilla prototype (can be seen as an "optimal" prototype for a given task), but obtaining it is not possible during the meta-test stage because the query set is not available. Hence, using the diffusion model to capture the distribution of how a vanilla prototype can be turned into an overfitted prototype is quite making sense.


**Weaknesses:**

- as the diffusion model is data-hungry, I believe when the number of available tasks is low and not diverse, this proposed model may not perform better than a conventional model. The author should also include this analysis to make the paper more complete.
- the proposed model has increased training time due to overfitting the prototype and a high number of timesteps to predict the prototype, as mentioned by the author. The author should also include this analysis to let readers know how much increased time over conventional methods.

**Questions:**

- I believe Eq.8 has a typo.
- What is the ratio of the increased training time over the conventional model?
- What is the case when the number of available tasks is low and not diverse? Will the conventional model perform better or not?
- How can this model be applied in production compared to conventional model?

**Limitations:**

- the author mentioned two limitations, one is the proposed model requires a substantial number of timesteps to sample the prototype during the meta-test stage.
- secondly, obtaining the overfitted prototype requires increased training time.

---

> ### Author Rebuttal · Authors · 2023-08-09
>
> ***Thank you for your positive feedback and for recommending acceptance of our paper.***
>
> * **Lower and not diverse available tasks**
>
> Answer: We acknowledge the reviewer's point regarding the data-intensive nature of diffusion models. Indeed, performance can be impacted when faced with a limited number of training or less diverse tasks.
>  Nevertheless, our original manuscript's Table 4 offers insights into cross-domain few-shot learning, indicating that even in less diverse scenarios, ProtoDiff maintains competitive results.
> Table 5 further illustrates our model's resilience under the constraints of the few-task few-shot learning framework. In addition, our approach consistently stands its ground against top-tier competitors, including MetaModulation (Sun et al.). We are confident these findings effectively address the issues highlighted and will provide more details in our revised version.
>
> |miniImageNet|1-shot|5-shot|
> |:-----:|:-----:|:-----:|
> |MetaModulation (Sun et al.)|43.21|57.26|
> |MLTI + ProtoDiff|44.75|58.18|
>
> Sun et al. "MetaModulation: Learning Variational Feature Hierarchies for Few-Shot Learning with Fewer Tasks." ICML 2023.
>
> * **Increased time over conventional methods**
>
> Answer: We appreciate your suggestion regarding the increased training time for the proposed model, which is a consequence of overfitting the prototype and the high number of timesteps required for predicting the prototype. To provide a clear comparison, we have analyzed the time consumed during both the meta-training and meta-testing phases of ProtoDiff and compared these results with the respective times for ProtoNet.
>  In meta-training and meta-testing wall-clock times per task, our ProtoDiff is slower by factors of  ProtoNet  $5 \times$ and $15 \times$, respectively. Numerical results are provided in our response to *Reviewer ZGD4*, who asked the same question.
> We will include this time analysis in the revised manuscript to understand our approach's efficiency comprehensively.
>
> * **Eq.8 typo**
>
> Answer: Correct, the summation symbol $\sum$ is missing over the task $T_i$. We have rectified this and will ensure a thorough review of all notations and equations in our manuscript to prevent any such oversights in the final draft.
>
> * **The ratio of the increased training time**
>
> Answer: The training time for our proposed model, ProtoDiff, compared to the conventional model, is approximately 2.5 times longer. This increase is primarily due to the overfitting of the prototype and the diffusion process to predict the prototype.
>
> * **The number of available tasks is low and not diverse**
>
> Answer: In scenarios where the number of tasks is limited and lacks diversity, the performance of conventional models might be questioned. However, in Table 4, our experiments focus on cross-domain few-shot classification and have already shown ProtoDiff's capabilities even when tasks are not diverse and exhibit domain shifts.
>  Further, our experiments in Table 5 delved into the few-task few-shot challenges, where the number of available tasks is low. In both these situations, ProtoDiff consistently emerges as one of the leading models, underscoring its effectiveness under such constraints.
>
> * **How can this model be applied in production?**
>
> Answer: Our ProtoDiff is designed to be adaptable in its application to any prototype-based meta-learning method, enhancing performance across different challenges, such as conventional few-shot learning, cross-domain few-shot learning, and few-task few-shot learning. The versatility of ProtoDiff allows it to handle various few-shot scenarios, providing a potential edge over standard methods. Such adaptability can be beneficial in diverse production environments.
>
> * **Limitations**
>
> Answer: We have included the comparisons for the effect of the number of timesteps (see response to *reviewer ZGD4*) and increased training time.

---

> > ### Comment · Reviewer_8tyC · 2023-08-15
> >
> > Thank you for the responses. I agree with the author's rebuttal.
> >
> > The only concern left for me is the increased training time which is an obvious tradeoff of this method. I believe this paper can inspire future work to reduce/improve this.
> >
> > I recommend an accept for this paper.

---

> > > ### Author Response · Authors · 2023-08-16
> > >
> > > Thank you for your positive feedback and recommendation to accept our paper. We acknowledge the concern regarding the increased training time, and we are motivated to explore more efficient implementations and methods in our future work, inspired by your insightful comments.

---

### Official Review · Reviewer_ZGD4 · 2023-07-06

**Soundness:** 3 good
**Presentation:** 3 good
**Contribution:** 2 fair
**Rating:** 6
**Confidence:** 5

**Summary:**

This paper proposes a prototype-based meta-learning method called ProtoDiff. During meta-learning stage, ProtoDiff introduces a task-guided diffusion process, learning a generative process conditioned on vanilla prototype for more robust prototype representation. ProtoDiff is extensively validated on within-domain, cross-domain, and few-task few-shot classification.



**Strengths:**

(1) The diffusion models have been very successful generative models, attracting great interests very recently. It is interesting to see the authors introduce the diffusion models into few-shot learning regime, for addressing the limitation of unrobust prototype estimation given a limited number of training examples.

(2) The proposed ProtoDiff is extensively evaluated on a variety of few-shot learning problems, including within-domain, cross-domain, and few-task few-shot classification. The ablation study is carefully and thoroughly designed for validating the effectiveness.


**Weaknesses:**

(1) The diffusion model for prototype learning is the key in the proposed ProtoDiff; however, some technical details on the diffusion model are missing.

About the architecture, how many layers are used in the encoder and what about dimension of linear transform and the number of heads in the attention module and what about the hidden dimensionality in MLP? What the proposed ProtoDiff will perform if the design parameters vary?

About the diffusion process, how do you select T? How does different T affect the performance and speed of the proposed ProtoDiff?

(2) Some other concerns.

In ProtoDiff, the ground truth prototypes, called overfitting prototypes, are obtained by finetuning the network for several steps. Besides that strategy, have you ever considered, during meta-training, using more support images (e.g. 8 or 16) by which one may get more accurate prototypes? As suggested in ProtoNet, it is beneficial to train with a higher way that will be used at meta-test.

In table 3, the baseline of ProtoNet is SetFeat; therefore, I suggest the authors change “This paper’’ to “SetFeat+ProtoNet”. How do you implement ProtoNet in Tables 4 and 5? Is it based on SetFeat or ProtoNet? Please clarify.

In equation (10), $\mathbf{z}$ is not defined; the subscript $t$ of $\sum$ is missing.



**Questions:**

See "Weaknesses" section.

**Limitations:**

Though the authors discuss the limitations of the proposed ProtoDiff, i.e., heavy computational cost, it is not clear how ProtoDiff is expensive compared to the baselines. I suggest the authors provide wall-clock time of meta-training/meta-testing of ProtoDiff against the ProtoNet.

---

> ### Author Rebuttal · Authors · 2023-08-09
>
> ***Thank you for your time, effort, and constructive comments.***
>
> (1) **Technical details are missing**
>
> Answer: Our ProtoDiff model is constructed using a design inspired by GPT-2. This includes a 12-layer transformer, a 512-dimensional linear transformation, an attention mechanism with 16 heads, and an MLP with a hidden dimensionality 512. These configurations are in line with GPT-2's default parameters. The configuration files can be accessed in our code repository for more detailed parameter setup. Our experiments in the subsequent tables highlight that our model achieves optimal performance with these settings. We will ensure this information is included in the revised manuscript for clarity.
>
>
> |transformer structures|1-shot|5-shot|
> |:-----:|:-----:|:-----:|
> |3-layer|62.17$\pm$ 0.25|78.93$\pm$0.17|
> |6-layer|63.25$\pm$ 0.22|79.63$\pm$0.15|
> |9-layer|65.21$\pm$ 0.21|80.13$\pm$0.18|
> |12-layer|66.63$\pm$ 0.21|83.48$\pm$0.15|
> |15-layer|64.15$\pm$ 0.25|80.93$\pm$0.18|
>
>
> |Heads numbers|1-shot|5-shot|
> |:-----:|:-----:|:-----:|
> |1|63.28$\pm$ 0.24|79.97$\pm$0.15|
> |4|64.23$\pm$ 0.22|80.13$\pm$0.14|
> |8|65.91$\pm$ 0.23|81.91$\pm$0.16|
> |16|66.63$\pm$ 0.21|83.48$\pm$0.15|
>
>  We have selected the diffusion time step **T** for the diffusion process to be 100. We've adopted the DDIM sampling strategy to accelerate prediction with $dim(\tau) = 10$. This effectively reduces the total sample requirement to just 10. We've conducted comparative experiments using varying diffusion times and intermediate intervals $dim(\tau)$. As presented in the subsequent tables, we observe that as the diffusion timesteps increase, both the performance and the inference time increase. Simultaneously, when $dim(\tau)$ is increased, the inference time decreases, making the process more efficient.
>
>
>
> |$dim(\tau)=10$, Timesteps|1-shot|5-shot|1-shot inference time|5-shot inference time|
> |:-----:|:-----:|:-----:|:-----:|:-----:|
> |T = 10|63.98$\pm$ 0.24|80.12$\pm$0.15|1ms|2ms|
> |T = 50|65.93$\pm$ 0.21|82.93$\pm$0.17|5ms|7ms|
> |T = 100|66.63$\pm$ 0.21|83.48$\pm$0.15|9ms|14ms|
> |T = 500|66.65$\pm$ 0.23|83.59$\pm$0.14|53ms|93ms|
> |T = 1000|66.74$\pm$ 0.20|83.97$\pm$0.17|102ms|192ms|
>
> |$dim(\tau)$, T=100|1-shot|5-shot|speed|
> |:-----:|:-----:|:-----:|:-----:|
> |1|66.78$\pm$ 0.21|83.62$\pm$0.14|99ms|192ms|
> |5|66.15$\pm$ 0.23|83.44$\pm$0.16|45ms|92ms|
> |10|66.63$\pm$ 0.21|83.48$\pm$0.15|9ms|14ms|
> |20|66.12$\pm$ 0.21|82.79$\pm$0.13|5ms|7ms|
> |100|63.15$\pm$ 0.23|81.27$\pm$0.14|1ms|2ms|
>
>
> (2) **More support images**
>
> Answer:  We appreciate your suggestion of using more support images during meta-training to obtain more accurate prototypes, as also recommended in ProtoNet. To shed more light on this, we've conducted an experimental comparison using different numbers of support sets during meta-training. The results in the subsequent table illustrate that augmenting the number of support images for each class during the meta-training phase enhances performance across various shots. We intend to integrate these findings into the revised version of our manuscript.
>
>
>
> |Number of support sets for each class|1-shot|5-shot|
> |:-----:|:-----:|:-----:|
> |1|66.63$\pm$ 0.21|73.12$\pm$0.15|
> |5|70.25$\pm$ 0.22|83.48$\pm$0.13|
> |8|73.91$\pm$ 0.21|84.72$\pm$0.13|
> |16|82.21$\pm$ 0.23|84.98$\pm$0.17|
>
> **Clarified Tables 4 and 5**
>
> Answer: Thank you for your keen suggestion. In our revision, we will change the label accordingly. For Table 4, the implementation is based on AFA [19]. Hence we will appropriately change "This paper" to "AFA + ProtoDiff". Similarly, in Table 5, the implementation is based on MLTI [51], and the label "This paper" will be replaced with "MLTI + ProtoDiff."
>
> **Equation typo**
>
> Answer:  Thank you for your careful observation. In Equation (10), **z** should be denoted as $\mathbf{z_t}$. Also, you're correct about the missing subscript for the summation symbol $\sum$. The subscript should be $(x, y) \sim Q$. Our revised manuscript will correct these typos to ensure clarity and precision.
>
> (3) **Limitations**
>
> Answer:  We have compiled the wall-clock time for both the meta-training and meta-testing phases of ProtoDiff and compared these against the respective times for ProtoNet. In meta-training and meta-testing wall-clock times per task, our ProtoDiff is slower by factors of  ProtoNet  $5 \times$ and $15 \times$, respectively.
> As part of future work, we will investigate and address this limitation to further enhance the efficiency of our approach.  We will incorporate these details into the revised manuscript for a more comprehensive comparison.
>
> |Meta-train|1-shot|5-shot|
> |:-----:|:-----:|:-----:|
> |ProtoNet|0.6ms|1.2ms|
> |ProtoDiff|3ms|5.3ms|
>
>
> |Meta-test|1-shot|5-shot|
> |:-----:|:-----:|:-----:|
> |ProtoNet|0.6ms|1.2ms|
> |ProtoDiff|9ms|14ms|

---

> > ### Comment · Reviewer_ZGD4 · 2023-08-16
> > **Reply to Rebuttal by Authors**
> >
> > I thank the authors for their careful responses, which addressed most of my concerns. I hope the authors include, in the modified version, the discussion and analysis herein about technical details and ablation on diffusion models, more accurate estimation of prototypes with more support images, and wall-clock time of meta-training/test. Despite the fact that the proposed ProtoDiff has heavy computational cost, I think it is interesting and inspiring to introduce diffusion model for robust prototype estimation in the few-shot learning. I opt to accept this paper.

---

> > > ### Author Response · Authors · 2023-08-16
> > >
> > > Thank you for your thoughtful review and your support for our work. We appreciate your suggestions and will definitely incorporate the detailed technical analysis, ablation on diffusion models, refined prototype estimation with additional support images, and the wall-clock time of meta-training/test in the revised version of the paper, thereby addressing the computational cost concern of ProtoDiff.

---

### Official Review · Reviewer_zHCT · 2023-07-07

**Soundness:** 3 good
**Presentation:** 2 fair
**Contribution:** 3 good
**Rating:** 7
**Confidence:** 5

**Summary:**

This paper tries to improve Prototypical Network in few-shot learning. Specifically, the authors use a diffusion model to predict optimal prototypes from initial prototypes. In each iteration of the meta-training phase, the optimal prototypes are obtained by several inner loops on the current task, then a diffusion model takes the initial prototypes as input, and predicts the difference between the optimal and initial prototypes. The final loss is used to meta-learn the feature extractor and the diffusion model. The experiments show that the proposed method can improve ProtoNet to some degree.

**Strengths:**

- Introducing diffusion models into few-shot learning and using it to translate features is interesting.
- The use of residual prototype learning is clever.

**Weaknesses:**

- In essence, the key idea is to learn a module to predict optimal prototypes from initial prototypes. Thus this module is a set-to-set function, and does not need to be stochastic, e.g., a MLP or transformer.  From this point, there is no motivation of using generative models like diffusion models, considering that no probabilistic inference is used during training or testing. The motivation of this paper is thus very weak.
- In fact, back in 2019, FEAT algorithm [1] already exploited such an idea, using a shallow transformer to transform initial prototypes into near-optimal prototypes without using any probabilistic models. While being much more efficient (no diffusion operations, no inner gradient loops, and much fewer parameters used) and being less tuned for hyperparameters, FEAT can achieve performance comparable to ProtoDiff.  Similarly, CrossTransformer [2] also uses a shallow transformer to transform initial prototypes into near-optimal prototypes, but in addition, considers spatial information. Thus the novelty and the value of the paper are quite limited.
- The writing of the method section needs to be significantly improved. Before looking at the algorithm 1-2 in the appendix, I struggle to understand the pipeline of the method, and some notations are confusing. For example, there are clear mistakes in the notations in lines 112-116.
- In Table 1, the result of Classifier-Baseline is borrowed from Meta-Baseline. Instead, the authors need to show the results of their own pre-trained model (trained with CE loss on the whole training set) before the use of ProtoDiff to show how much improvement of their method brings.
- In Table 2, only generative models are considered. As stated above, more deterministic feature transformation methods should be compared.

[1] Few-Shot Learning via Embedding Adaptation with Set-to-Set Functions. CVPR 2020.
[2] CrossTransformers: spatially-aware few-shot transfer. NeurIPS 2020.

======Post-rebuttal=======

The authors have addressed most of the concerns during the rebuttal. Thus I raise my score from 3->7.

**Questions:**

See Weakness above.

**Limitations:**

The authors have stated some of the limitations.

---

> ### Author Rebuttal · Authors · 2023-08-09
>
> ***Thank you for your time, effort, and constructive comments.***
>
> * **Motivation of ProtoDiff**
>
> Answer: We acknowledge your insight regarding using a stochastic or diffusion model. Our choice to employ a generative model like the diffusion process extends beyond simply predicting optimal prototypes from initial ones. It also incorporates uncertainty into the few-shot learning process, which can be vital when dealing with limited samples.  We have also conducted the experiments only using MLP and transformers instead of using generative models in the following table.
>
> |miniImageNet|1-shot|5-shot|
> |:-----:|:-----:|:-----:|
> |Vanilla|63.17|79.26|
> |MLP|64.15|80.23|
> |Transformer|64.97|81.28|
> |Diffusion|66.63|83.48|
>
> While an MLP or transformer can improve performance over vanilla prototypes, our experiments reveal that ProtoDiff outperforms these alternatives. Incorporating uncertainty through the generative model provides a nuanced advantage in capturing the underlying distribution, resulting in better performance. We will detail these experimental comparisons and further clarify the motivation behind our approach in the revised manuscript.
>
> * **Limited novelty**
>
> Answer: While FEAT and CrossTransformer also utilize a function to transform vanilla prototypes into near-optimal ones, they lack the intermediate supervision provided by "overfitted" prototypes, which can guide the transformed prototype toward an optimal state. Our model uniquely learns an "overfitted" prototype first and then leverages a diffusion model to capture this learning process, ensuring a more refined transition to the optimal prototype. Besides the diffusion process to generate diffused prototypes, our model provides a new insight to leverage the “overfitted” prototype as supervision. This approach enables our system to utilize the overfitted prototype as guidance, thus implementing meta-learning as a continuous learning process.  We will incorporate these papers into our discussion of related work and in our comparisons. We believe this is of community interest.
>
> * **Writing of the method**
>
> Answer: We regret that the notations in the method section were confusing, and we appreciate pointing out the specific mistakes in lines 112-116. We have taken your feedback to heart, fixed these typos, and conducted a thorough review of all notations and equations in the manuscript.
>
> * **Pre-trained model**
>
> Answer: We agree that it is essential to showcase the results of our own pre-trained model before applying ProtoDiff to provide a clear comparison and demonstrate our method's improvements. In response to your suggestion, we have prepared the results of our pre-trained model (trained with CE loss on the whole training set) and will present it in a new table. This information will be included in the revised manuscript to provide a more transparent and comprehensive evaluation of ProtoDiff.
>
> |||mini  1-shot|mini   5-shot|tiered 1-shot|tiered  5-shot|800  1-shot|800  5-shot|
> |:-----:|:-----:|:-----:|:-----:|:-----:|:-----:|:-----:|:-----:|
> |Pretrained from Chen et al.|Classifier-Baseline|58.91|77.76|68.07|83.74|86.07|96.14|
> |Pretrained from Chen et al.|ProtoDiff|66.63|83.48|72.95|85.15|92.13|98.21|
> |Pretrained from ours|Classifier-Baseline|58.75|77.86|68.15|83.95|85.97|96.03|
> |Pretrained from ours|ProtoDiff|66.49|83.51|73.01|85.72|92.05|98.11|
>
> * **More deterministic feature transformation methods results**
>
>  We have also conducted the experiments using MLP and transformers instead of generative models in the first question (**Motivation of ProtoDiff**).
> While an MLP or transformer can improve performance over vanilla prototypes, our experiments reveal that ProtoDiff outperforms these alternatives.

---

> > ### Comment · Reviewer_zHCT · 2023-08-10
> > **Response to Authors Rebuttal**
> >
> > Thanks for the rebuttal. I totally agree that uncertainty is vital when dealing with limited samples. However, there is no "useful" uncertainty in this paper due to design choices. Let me detail it below.
> >
> > When uncertainty is taken into account, only when there are multiple samples sampled from the posterior probability which are used for inference, the uncertainty makes sense (as what is done in variational inference). However, in this paper, during inference, only one prototype is produced for each class, thus there is no probabilistic inference, i.e., the distribution built for the prototype makes no use at all and, the uncertainly is not taken into account; furthermore, it can be easily inferred that the optimal prototype becomes the best choice if only one prototype is produced. So in principle, in the authors' method, the optimal posterior prototype distribution should be a point estimate of the optimal one, so there is, in fact, no uncertainty in your method. The strong performance only comes from the auxiliary deep architecture (used for the diffusion process) for estimating the optimal prototypes, which does not need uncertainty at all.
> >
> > Besides, your baseline using transformer is far below that reported by FEAT [1], the method identical to adding a transformer layer for estimating the optimal prototype. I have re-implemented FEAT, and it can exactly match the reported performance, which is very similar to ProtoDiff. This indicates that generative modeling used in this paper is not needed, at least when used without probabilistic inference.
> >
> > I know that different from FEAT, Protodiff learns to fit the overfitted prototype explicitly. However, due to the mechanism of meta-learning, FEAT implicitly does such a process (meta-learn to map prototypes to optimal ones, the "supervision/guidance" in your rebuttal), and its performance matches Protodiff closely. I think it is interesting to take this mapping process out explicitly, but I still do not think there is anything new here.
> >
> > [1] Few-Shot Learning via Embedding Adaptation with Set-to-Set Functions. CVPR 2020

---

> > > ### Author Response · Authors · 2023-08-14
> > > **Thank you for the response.**
> > >
> > > We thank the reviewer for engagement and encouragement to sharpen our contribution and its justification.
> > >
> > > **Uncertainty in diffusion and ProtoDiff**: Traditional diffusion models, during the sampling phase, introduce randomness at every time step by drawing random noise from a standard normal distribution to diffuse the sample. This mechanism injects uncertainty into the diffusion process at each step (full details in Jonathon Ho et al. Denoising Diffusion Probabilistic Models. NeurIPS 2020.) Our ProtoDiff follows a similar strategy. At each time step, we sample different random noise values to diffuse our prototypes, ensuring that uncertainty is introduced in each step of the prototype generation.
> > > While it is true that we generate a single prototype for each class during inference, it's important to note that the inherent uncertainty introduced during the diffusion process ensures robustness and generalization in this prototype (see Figure 7). Multiple samples at the final step could indeed introduce additional uncertainty. However, we believe the cumulative uncertainty introduced during each step of the diffusion process suffices, and further sampling might increase the computational complexity without a proportionate gain in performance.
> > >
> > >
> > > **Differences and complementarity with FEAT**:  Thank you for the insightful comment. Allow us to clarify some misunderstandings and highlight the difference in our approach compared to FEAT.
> > > The way our Transformer and ProtoDiff construct a new prototype is fundamentally different from FEAT. FEAT leverages samples from *all categories* within a task to generate a prototype, capitalizing on the intrinsic inter-class information to derive a more discriminative prototype. In contrast, our ProtoDiff, while using the overfitted prototype as supervision, only employs samples from a *single category* to generate the new prototype. This means we are not tapping into the potential informative context provided by samples from other categories. Additionally, our ProtoDiff employs the diffusion model to progressively generate the prototype, whereas FEAT does not utilize any generative model for prototype creation.
> > > Naturally, we may adopt FEAT’s strategy of utilizing samples from all categories to produce a new prototype. The results with ResNet-12, as presented in the following table, are revealing. Our performance with a transformer based on an overfitted prototype (with all categories) indeed experiences a slight enhancement over FEAT. However, when the diffusion process is integrated, there is a notable improvement of 2.19% and 3.11%  in the 1-shot and 5-shot scenarios on miniImageNet. We will include these results in the main paper.
> > >
> > >
> > > |miniImageNet|1-shot|5-shot|
> > > |:-----:|:-----:|:-----:|
> > > |ProtoNet|63.17|79.26|
> > > |FEAT|66.78|82.05|
> > > |FEAT (Our reimplementation)|66.93|82.41|
> > > |FEAT + overfitted prototype|67.68|83.18|
> > > |FEAT + ProtoDiff|68.97|85.16|
> > >
> > >
> > > We believe our approach, while drawing parallels with FEAT, introduces unique considerations in generative prototype creation. We hope this response clarifies our position as well as the novelty of our contributions.

---

> > > > ### Comment · Reviewer_zHCT · 2023-08-15
> > > >
> > > > Thanks for the additional response. I increase my score from 3 to 4 given that the method can indeed improve on FEAT, showing some differences there. The reason why I cannot give a higher score is that the authors still do not clarify the necessity of the generative module in the paper. The authors say that “the inherent uncertainty introduced during the diffusion process ensures robustness and generalization in this prototype (see Figure 7)”. However, Figure 7 only shows the benefit of transforming the prototype to the overfitted one, not the benefit of uncertainty. As I stated before, **uncertainty is useful only when the uncertainty itself is utilized for prediction. Multiple prototypes sampled from the distribution make sure that the prediction won't be biased due to a single overconfident estimation of prototype, thus being more robust and generalizable**. An ideal experiment would be sampling several diffused prototypes and predicting through some probabilistic inference methods (e.g., MAP, variational inference, bayesian ensemble), like what has been done in Bayesian meta-learning literature; see [1-2]. Only if such a probabilistic inference process improves performance, the uncertainty makes sense.
> > > >
> > > > [1] Bayesian Model-Agnostic Meta-Learning. NeurIPS 2018.
> > > >
> > > > [2] Amortized Bayesian Meta-Learning. ICLR 2019.

---

> > > > > ### Author Response · Authors · 2023-08-17
> > > > >
> > > > > Thank you for increasing your score and guiding us further with actionable suggestions to exploit the inherent uncertainty in our approach more effectively.
> > > > > In alignment with the amortized inference approach of variational autoencoders and the methods outlined in provided references [1-2], we conducted the experiments you suggested with FEAT + ProtoDiff and Meta-Baseline + ProtoDiff, where we sampled multiple prototypes in the final step. The results of this experiment are illustrated in the following tables. We observed a considerable improvement in performance as we increased the number of samples.  A sampling size of 50 yields remarkable results, e.g., increasing 1-shot accuracy from 66.63 (for meta-baseline + ProtoDiff) to 72.25 (for FEAT + ProtoDiff). We are genuinely excited by this principled improvement and are grateful for your suggestion that led us to explore this aspect. The increased inference time for 50 samples is noticeable but manageable, and we believe the considerable performance improvement justifies it.
> > > > >
> > > > >
> > > > > |FEAT + ProtoDiff|1-shot|5-shot|1-shot Inference time|5-shot Inference time|
> > > > > |:-----:|:-----:|:-----:|:-----:|:-----:|
> > > > > |1 |68.97|85.16|10ms|14ms|
> > > > > |10|70.18|86.53|12ms|17ms|
> > > > > |20|71.07|87.14|13ms|19ms|
> > > > > |50|72.25 |88.32 | 16ms|22ms|
> > > > > |100|72.21 |88.37 | 23ms|37ms|
> > > > > |1000|72.12 |88.13| 45ms|54ms|
> > > > >
> > > > >
> > > > >
> > > > > |Meta-Baseline + ProtoDiff|1-shot|5-shot|1-shot Inference time|5-shot Inference time|
> > > > > |:-----:|:-----:|:-----:|:-----:|:-----:|
> > > > > |1 |66.63|83.48|9ms|14ms|
> > > > > |10|68.02|84.95|12ms|16ms|
> > > > > |20|68.91|85.74|13ms|17ms|
> > > > > |50|69.14 |86.12 | 14ms|21ms|
> > > > > |100|69.13 |86.07 | 21ms|35ms|
> > > > > |1000|69.11 |86.11| 43ms|52ms|
> > > > >
> > > > > We will incorporate these additional experiments and findings into the final version of our manuscript. We will add a closing discussion on the balance of uncertainty and computation cost of our approach, which hints at a worthwhile endeavor for future works. Thank you.

---

> > > > > > ### Comment · Reviewer_zHCT · 2023-08-17
> > > > > >
> > > > > > I am really impressed by the performance! This clearly shows the benefit of the uncertainty modeled during training. I thus raise my score to 7 (accept). I highly encourage the authors to add all the above discussions to the camera-ready version if accepted.

---

> > > > > > > ### Author Response · Authors · 2023-08-17
> > > > > > >
> > > > > > > Thank you very much for your kind words and for raising your score to an acceptance level. We are thrilled that our work and the efforts to model uncertainty during training have impressed you. We absolutely agree with your suggestion and are committed to incorporating all the valuable discussions, including the comparison with FEAT's implementation and the extensive discussions on model uncertainty, into the camera-ready version of our paper. This will ensure that the final manuscript provides a clearer and more comprehensive understanding of our work. Thank you again for your insightful feedback and encouragement throughout this process.

---

### Decision · Program_Chairs · 2023-09-21

**Decision:**

Accept (poster)

**Comment:**

This paper proposes a prototype-based meta-learning method, ProtoDiff.  It designs a task-guided diffusion process to predict the more optimal prototypes from initial prototypes. Experimental results show the effectiveness of ProtoDiff on within-domain, cross-domain, and few-task few-shot classification tasks. All the reviewers unanimously accepted the paper, and so did the final decision. The authors should also revise their paper according to the reviewers' suggestions in the final version.